# Weak-to-Strong Generalization under Distribution Shifts

**Myeongho Jeon**[1,*]  **Jan Sobotka**[1,*]  **Suhwan Choi**[2,*]  **Maria Brbić**[1,†]
[1]EPFL   [2]Seoul National University

## Abstract

As future superhuman models become increasingly complex, accurately supervising their behavior may exceed human capabilities. Recent works have demonstrated that in such scenarios, weak models can effectively supervise strong models, a phenomenon known as weak-to-strong generalization. However, we find that naive weak-to-strong generalization fails under distribution shifts, often leading to worse performance of the strong model than its weak supervisors. To address this, we propose RAVEN, a robust weak-to-strong generalization framework that dynamically learns the optimal combinations of weak models in addition to parameters of the strong model. We demonstrate the effectiveness of RAVEN on image classification, text classification, and preference alignment tasks. RAVEN outperforms alternative baselines by over $30\%$ on out-of-distribution tasks while matching or surpassing existing methods on in-distribution tasks. Moreover, our results show that RAVEN assigns higher weights to more accurate weak models, demonstrating its ability to automatically identify trustworthy supervision.

## 1   Introduction

Recent AI systems have reached near-human performance through extensive pre-training on large datasets, followed by fine-tuning with human supervision. Techniques such as supervised fine-tuning, reinforcement learning with human feedback (RLHF) [17, 61, 50], and direct preference optimization (DPO) [43, 56] are key examples of effectively aligning models with human preferences. A fundamental assumption in these methods is that the human supervision is of high quality. However, when the data to be annotated is beyond human comprehension, providing reliable supervision becomes challenging. For instance, in domains such as cosmology, healthcare, or biology, even experts may struggle with accurate labeling. Furthermore, during the alignment phase with human preferences, if a superhuman-level model generates highly complex outputs, humans may struggle to fully understand them, making it challenging to provide effective feedback. This challenge is known as *weak supervision* [10].

So, how can humans supervise a superhuman model that surpasses their own capabilities? To mimic this future scenario, an analogous framework has been proposed in [10], where a weak model simulates human supervision, while a strong model acts as a proxy for a potential superhuman model. In this setup, weak supervision signals are utilized to train the strong model, aiming to surpass the weak model's performance and approach the performance achieved when trained with ground-truth (GT) labels. Conceptually, this approach, termed weak-to-strong (W2S) generalization [10], could lead toward developing superhuman models, as surpassing the performance of a weak model in this context could eventually mean exceeding human capabilities. Moreover, such success would be practical even before superhuman models emerge—for instance, aligning GPT-5 using only GPT-4-level supervision could simplify model alignment today [10]. Recent works [10, 26, 45, 19]

---

*Equal contribution
†Correspondence to: mbrbic@epfl.ch

39th Conference on Neural Information Processing Systems (NeurIPS 2025).

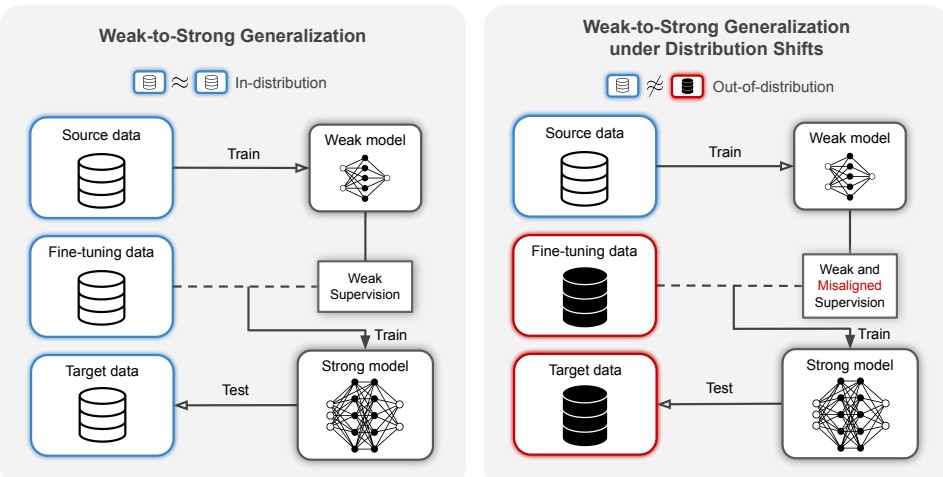

Figure 1: An illustration of weak-to-strong generalization under distribution shifts. *Left*: The original W2S generalization framework [10]. *Right*: Our extended framework that incorporates distribution shift. In this scenario, the weak model is not only limited in understanding but also *unfamiliar* with the fine-tuning distribution, making it less reliable for annotating data to supervise the strong model.

have shown that W2S generalization is indeed *feasible*, meaning that strong pre-trained models naturally generalize beyond their weak supervisors.

However, humans may encounter data that is not only complex but also unfamiliar, making the weak supervision even weaker. For example, a radiologist used to one type of imaging data may misinterpret scans from a different machine or mislabel rare diseases outside their usual clinical practice [8]. As a result, expert annotations can become even less reliable due to unfamiliarity with the data. To simulate this, we consider a scenario in which the weak model is trained on data drawn from a distribution that differs significantly from the strong model's fine-tuning data, a setting we refer to as *weak and misaligned supervision* (Figure 1). This raises the question: Is W2S generalization still feasible under distribution shifts?

Following this question, we interestingly find that weak supervision becomes significantly less effective in out-of-distribution (OOD) settings compared to in-distribution (InD) settings and, in some cases, *even not feasible*, leading to the strong model performing worse than the weak model. This is a nontrivial issue, as it suggests that unnoticed misaligned human perspectives can affect the annotation process and potentially reduce performance. These findings motivate the need for a robust W2S generalization framework that remains effective under distribution shifts.

Here, we propose *Robust AdaptiVe wEightiNg* (**RAVEN**)[3], a robust W2S generalization framework in which the strong model dynamically learns to combine the outputs from an ensemble of weak annotators. Under distribution shift, weak models exhibit higher variance in performance compared to the InD setting. RAVEN mitigates this by learning to assign higher weights to more reliable weak models. Specifically, the supervision weights assigned to the weak models are updated iteratively and jointly trained with the strong model's parameters. This approach effectively handles high variance of weak models by allowing the strong model to prioritize the most favorable weak model for the fine-tuning data distribution. Additionally, to guide the strong model to learn how to choose the reliable weak model in the early stages of training, RAVEN introduces *easy-sample guided initialization*, which trains the strong model exclusively on samples where the weak models consistently provide the same predictions.

We evaluate RAVEN on image classification, text classification, and preference alignment in text generation tasks. RAVEN achieves a $55\%$ improvement in image classification, a $57\%$ improvement in text classification, and a $33\%$ improvement in preference alignment compared to the best alternative baselines for each task. Remarkably, although the information about the performance of weak models is unknown to the strong model, we observe that the strong model typically assigns the highest weight to the best-performing weak model without any additional guidance.

---

[3]Project website with code: `https://brbiclab.epfl.ch/projects/raven`

## 2 Problem statement

**Weak-to-strong generalization.** In W2S generalization setting [10], a strong (large) pretrained model is fine-tuned using labels generated by a weak (small) model, denoted by $f_s$ and $f_w$, respectively. Given input space $X$ and label space $Y$, the data is divided into source $P_{src}(X, Y)$, fine-tuning $P_{tuning}(X, Y)$, and target data $P_{trg}(X, Y)$. The following procedure is then evaluated: *(i)* **Generate a weak supervisor**: $f_w$ is trained on $P_{src}$ with ground-truth (GT) labels, denoted by $f_w^{src}$. *(ii)* **Train a strong model with weak supervision**: $f_s$ is trained on $P_{tuning}$ using weak pseudo-labels generated by $f_w^{src}$, denoted by $f_s^{pseudo}$. *(iii)* **Train a strong model with GT as a ceiling**: $f_s$ is trained on $P_{tuning}$ with GT, denoted by $f_s^{gt}$. The performance gap recovered (PGR) on $P_{trg}$ is then calculated as follows:

$$PGR := \frac{Acc(P_{trg}; f_s^{pseudo}) - Acc(P_{trg}; f_w^{src})}{Acc(P_{trg}; f_s^{gt}) - Acc(P_{trg}; f_w^{src})}, \tag{1}$$

where $Acc(A; B)$ denotes the accuracy of model B on data A. The goal of W2S generalization is to achieve a PGR close to 1 by making the target model $f_s^{pseudo}$ approximate $f_s^{gt}$, indicating that the model reaches ground-truth-level performance even when trained with weak supervision. In [10], the assumption is that $P_{src}$, $P_{tuning}$, and $P_{trg}$ are drawn from the same underlying distribution.

**Weak-to-strong generalization under distribution shifts.** We extend the concept of W2S generalization by considering distribution shifts between $P_{src}$ and $P_{tuning}$. We define distribution shift through the generalization gap: $\Delta R(f) = R_{tuning}(f) - R_{src}(f)$, where $R(f) = \mathbb{E}_{(x,y) \sim P}[\mathcal{L}(f(x), y)]$ is the expected risk, $f$ denotes the model, and $\mathcal{L}$ is the loss function. A large $\Delta R(f)$ indicates substantial distribution shift from $P_{src}$ to $P_{tuning}$, which can make weak supervision misaligned[4]. This conceptualizes the case where humans annotate difficult-to-understand and unfamiliar domains, leading to weak and misaligned supervision. This could, for example, happen with medical data that are both highly specialized and cross-institutional. In this setting, PGR, as defined in Eq. (1), evaluates how effectively weak supervision under distribution shift can be leveraged to enhance the capabilities of the strong model.

**Observation.** To evaluate the robustness of naive W2S generalization under distribution shifts, we analyze changes in PGR that happen in OOD scenarios. Specifically, we introduce a shift from IMAGENET to IMAGENET-C, treating 19 corruption types as distinct domains. For comparison, we establish both InD and OOD setups. In both cases, we use the IMAGENET training set as $P_{src}$, while the validation set from IMAGENET serves as $P_{tuning}$ and $P_{trg}$ for the InD scenario, and IMAGENET-C serves as $P_{tuning}$ and $P_{trg}$ for the OOD scenario.

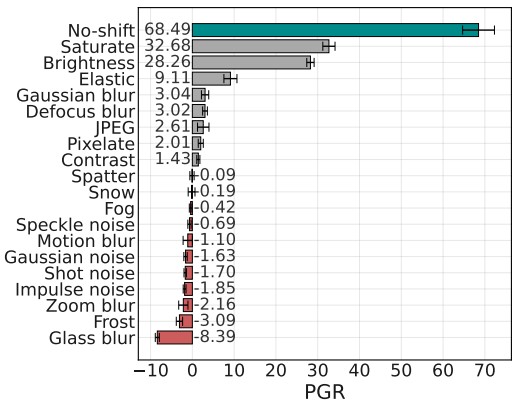

We find that OOD generalization rarely leverages the strong model's capabilities as much as when no distribution shift is present (Figure 2). Surprisingly, on 11 out of 19 IMAGENET-C corruption cases, the W2S generalization is even infeasible, *i.e.*, the W2S performance is worse than that of the weak model. This shows the need for an approach that achieves a robust W2S generalization under distribution shifts.

Figure 2: Performance variation with distribution shifts. The Cyan bar indicates positive PGR of InD, gray represents positive PGR for OOD, and red denotes negative PGR. PGR is reported as a percentage. Detailed results can be found in Appendix B.

## 3 Preliminary

**WeakS-to-strong generalization.** Inspired by the concept that multiple experts can collectively provide effective supervision even if each individual expert is insufficient on its own, weakS-to-strong framework [19] utilizes multiple weak models for training. A straightforward approach to ensemble

---

[4]A more detailed formal definition of distribution shift is provided in Appendix A.

the weak models is to use a weighted sum to combine their predictions, followed by calculating the loss:

$$\mathcal{L}_{\text{ensemble}}(\Theta_{\text{S}}, \mathbf{W}) := \mathcal{L}_{\text{CE}}(f_s(x), \sum_{i=1}^{M} w^{(i)} f_{w_i}^{src}(x)), \tag{2}$$

where $\mathbf{W} := \{w^{(i)}\}_{i=1}^{M}$ denotes pre-defined weights set in ensembling (*e.g.*, the average) $M$ weak models and $\mathcal{L}_{\text{CE}}$ denotes cross-entropy loss. $f_{w_i}$ denotes $i$-th weak model and $\Theta_{\text{S}}$ represents the parameters of the classifier in the strong model $f_s$.

## 4 RAVEN: Robust adaptive weighting approach

In W2S generalization under distribution shift, the challenge is that weak models are trained on data distributions that are different from the strong model's fine-tuning data distribution, making them even less reliable. Thus, leveraging an ensemble of weak models in Eq. (2) becomes an effective strategy. However, compared to the InD setting, weak models have substantially higher performance variance in OOD scenarios (Figure 3), and their effectiveness varies considerably across domains (Appendix H.3). Similarly, in an analogous real-world scenario, some human annotators may adapt better than others to unfamiliar domains. Consequently, it is crucial to *identify more reliable weak models within the ensemble*. This motivates the need for a dynamic selection mechanism that selects the most suitable weak model(s) for a given fine-tuning dataset $P_{tuning}$.

Motivated by this, we propose **RAVEN**, a robust W2S generalization framework that dynamically learns how to combine different weak models. To improve ensemble performance in OOD settings, RAVEN incorporates two key components: *(i) adaptive weighting*, and *(ii) easy-sample guided initialization*.

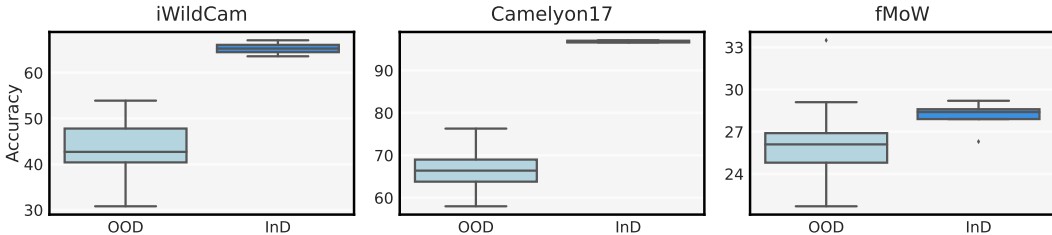

Figure 3: Across different datasets, we observe substantially higher performance variance in OOD scenarios than in InD scenarios. The variance is computed using 20 weak models that were initialized with different random seeds and trained on datasets from the WILDS benchmark [35].

**Adaptive weighting.** Among the weak models, some are more reliable than others. The key idea in RAVEN is to prioritize these models' predictions by assigning different weights to their weak supervisions during the strong model's training. We begin by training $M$ weak models with different random seeds, which represent human annotators with diverse backgrounds. This approach is inspired by the findings of [40, 37], which demonstrate that training individual networks with random initialization is often sufficient to achieve diversity in practice. In addition to achieving diversity via different random seeds, we also explore using different architectures and data sources when training weak models (Section 5.3). Leveraging these diverse weak models, we introduce adaptive weighting loss for the W2S training:

$$\mathcal{L}_{\text{adaptation}}(\Theta_{\text{S}}, \Theta_{\text{W}}) := \mathcal{L}_{\text{CE}}\left(f_s(x), \sum_{i=1}^{M} \theta^{(i)} f_{w_i}^{\text{src}}(x)\right) \quad \text{s.t.} \sum_{i=1}^{M} \theta^{(i)} = 1, \ \theta^{(i)} \in \Theta_{\text{W}}. \tag{3}$$

where $\Theta_{\text{W}} := \{\theta^{(i)}\}_{i=1}^{M}$ represents the weights used to linearly combine the outputs of the weak models. Importantly, the weak models themselves are fixed, while the ensembling weights $\{\theta^{(i)}\}_{i=1}^{M}$ are fully trainable. At each step, these weights are adjusted, leading to a change in how the weak models are utilized throughout training. With this objective, we train a linear classifier parameterized by $\Theta_S$ on top of the frozen pre-trained backbone of the strong model. The backbone generates robust representations, serving as an effective anchor to train the linear classifier using the weak models.

**Easy-sample guided initialization.** To minimize the objective, the strong model may resort to shortcutting by assigning excessive weight to the weak model most similar to its initial classifier. This behavior is especially problematic during early training when the strong model's classifier is still under-optimized and performs poorly. To address this, we implement a warm-up strategy for the strong model's classifier using easy samples—those on which all the weak models agree, *i.e.*, give the same predictions. During this phase, the weights $\theta^{(i)} \in \Theta_W$ are fixed to $1/M$. Afterward, we enable adaptive weighting across all samples, transitioning from $\mathcal{L}_{\text{ensemble}}$ with easy samples to $\mathcal{L}_{\text{adaptation}}$ for all the samples. The concept of initializing training with easy samples has proven effective for learning with noisy labels [15]. This approach aligns with W2S generalization, as weak supervision often involves noisy labels resulting from incorrect predictions.

**Optimization procedure.** After the easy-sample guided initialization, the loss for adaptive weighting $\mathcal{L}_{\text{adaptation}}$ is optimized by alternating updates to $\Theta_S$ and $\Theta_W$ at each step, as follows:

$$\Theta_S^* := \arg\min_{\Theta_S} \mathcal{L}_{\text{adaptation}}(\Theta_S, \Theta_W^*), \quad \Theta_W^* := \arg\min_{\Theta_W} \mathcal{L}_{\text{adaptation}}(\Theta_S^*, \Theta_W). \tag{4}$$

This alternating optimization effectively balances the trade-off between optimizing the strong model and leveraging weak models. The overall RAVEN training procedure is summarized in Algorithm 1.

---

**Algorithm 1** Robust Adaptive Weighting (RAVEN)

---

**Require:** Fine-tuning dataset $D_{tuning}$, weak models $\{f_{w_i}^{\text{src}}\}_{i=1}^M$, pretrained strong backbone
**Ensure:** Trained linear classifier $\Theta_S$
 1: **Easy-sample guided initialization:**
 2: Identify $D_{\text{easy}} = \{x \in D_{tuning} \mid \arg\max_k f_{w_i}^{\text{src}}(x)[k] = \arg\max_k f_{w_j}^{\text{src}}(x)[k], \ \forall i, j\}$
 3: Fix $\theta^{(i)} = \frac{1}{M}$ and define $\Theta_W := \{\theta^{(i)}\}_{i=1}^M$, with $\sum_{i=1}^M \theta^{(i)} = 1$
 4: Warm up $\Theta_S$ on $D_{\text{easy}}$ by minimizing $\mathcal{L}_{\text{ensemble}}$ (2)
 5: **Adaptive weighting on the full dataset:**
 6: **while** not converged **do**
 7:     Update weights: $\Theta_W \leftarrow \arg\min_{\Theta_W} \mathcal{L}_{\text{adaptation}}(\Theta_S, \Theta_W), \quad x \in D_{tuning}$ (3)
 8:     Update strong model: $\Theta_S \leftarrow \arg\min_{\Theta_S} \mathcal{L}_{\text{adaptation}}(\Theta_S, \Theta_W), \quad x \in D_{tuning}$ (3)
 9: **end while**
10: **return** $\Theta_S$

---

*Remark* 1. We observe that, in most cases, the strong model assigns the highest weight $\max_i \{\theta^{(i)}\}_{i=1}^M$ to the best-performing weak model for $P_{trg}$ by the end of training, without requiring additional guidance. This behavior enhances robustness, as weak models often exhibit significant variation in OOD performance, and there is a strong positive correlation between weak and W2S performance. This capability is particularly advantageous because it allows for the automatic identification and utilization of the best weak model in terms of $P_{trg}$, even when the GT labels for the target data are unknown, and the best-performing weak model is therefore unclear. We provide a theoretical analysis of this property of RAVEN in Appendix C, along with detailed quantitative results in Section 5.3, Appendix H.4, and Appendix H.5.

## 5 Experiments

We evaluate RAVEN on image classification, text classification, and preference alignment in text generation tasks. Classification training involves two stages: pre-training and fine-tuning. While pre-training is performed in a self-supervised or unsupervised manner [14, 12, 55], fine-tuning typically relies on human annotations. In our approach, we focus on fine-tuning by replacing human annotations with predictions of weak models. For text generation tasks, pre-training, supervised fine-tuning, and human preference alignment are the conventional learning phases for foundation models. In this work, we specifically focus on alignment, substituting human feedback with preference predictions from weak models.

## 5.1 Experimental setup

**Image classification.** For OOD setting, we use IWILDCAM [5], CAMELYON17 [39], and FMOW [18] as benchmarks to evaluate our framework. In these datasets, the domain is defined by the location of the camera, the hospital, and the time, respectively. For each dataset, the training set is used as the source data $P_{src}$, 70% of the OOD validation set is randomly selected as fine-tuning data $P_{tuning}$, 10% is reserved as the validation set for hyperparameter tuning, and the remaining 20% is designated as the target data $P_{trg}$. For the InD scenario, we adopt the same approach as [10], which utilized IMAGENET. Consistent with the setup in [10], we use AlexNet as the weak model and DINO ViT8/B as the strong model.

RAVEN is compared to several W2S approaches, including Naive weak-to-strong (Naive) [10], weakS-to-strong with uniform weights (Ens) (Eq. (2)), Auxiliary confidence loss (Conf) [10], Bootstrapping (Boots) [10], Vision superalignment (V-sup) [26], Bayesian weakS-to-strong (Bayes) [19], and Co-supervised learning (Co-sup) [45][5].

**Text classification.** We employ AMAZON-WILDS [35], MEDMCQA [51], and MEDQA [31] to evaluate RAVEN for text classification. AMAZON-WILDS, a sentiment analysis dataset, exhibits both domain and subpopulation shifts. In this dataset, domains correspond to individual reviewers. For the InD W2S scenario, the InD validation data is used in place of the OOD validation set. We designate Llama-3.2-1B [24], Qwen2.5-0.5B [65] as the weak models and Llama-3.1-8B, Qwen2.5-7B, Qwen2.5-14B as the strong models. For the medical benchmarks, we adopt MEDMCQA as the source dataset and MEDQA as the fine-tuning and target datasets. These originate from different examination systems from India and the U.S. [6], resulting in a natural domain shift between them. For these medical benchmarks, we use Qwen-2.5-0.5B as a weak model and Meditron-7B and Meditron-70B [16] as strong models.

**Preference alignment.** While Burns et al. [10] highlighted the importance of alignment in W2S, their focus was solely on maximizing preference prediction accuracy rather than exploring alignment itself. Cui et al. [19] investigated the alignment phase but limited their focus to slot filling tasks. In contrast, we assess preference alignment in the context of text generation on HH-RLHF [4], OPENAI SUMMARIZE FROM FEEDBACK[61], and HUMAN-LIKE DPO[11] datasets.

For HH-RLHF, we create a distribution shift by using Helpfulness samples for $P_{src}$, randomly sampled Harmlessness samples for $P_{tuning}$, and 1,000 Harmlessness samples for $P_{trg}$; for InD setting, we use only Harmlessness samples. In the second setup, we use OPENAI SUMMARIZE FROM FEEDBACK as $P_{src}$ and HUMAN-LIKE DPO as $P_{tuning}$. To align the strong model with $P_{tuning}$ and the preference predictions of the weak models (*i.e.*, feedback), we employ DPO [56]. To integrate RAVEN into alignment tasks, we propose a novel objective, **DPO-R** (Appendix E.2). We use Qwen2.5-0.5B and Qwen2.5-7B as the weak and strong models, respectively. Following Rafailov et al. [56], we use the `GPT-4o win rate` (WR) as our evaluation metric (Appendix E.4), and compare RAVEN to Naive, Ens, and Bayes baselines (details in Appendix E). Conf and V-sup baselines are not applicable due to the misalignment of outputs between the weak and strong models, and Boots and Co-sup baselines are unsuitable due to task differences.

**Experimental details.** For each experiment, we perform a grid search to select the learning rate (including its initial value and decay schedule) and the number of training iterations based on validation loss. To determine the duration of the easy-sample guided initialization phase, we conduct a grid search over $\{10\%, 20\%, 50\%\}$ of the total iterations. Additional implementation details for the method and evaluation protocol are provided in the Appendix D.

## 5.2 Experimental results

**Image classification.** Results on the image classification task show that RAVEN consistently outperforms all the baselines in the W2S generalization under distribution shift scenario (Table 1). RAVEN significantly enhances robustness in the OOD setting, while also surpassing other baselines

---

[5]We adopt the domain generalization method outlined in Section 5.2 of [45] for robust W2S experiments, rather than their original approach, as it is better suited to the robust W2S scenario. Further details can be found in Appendix D.5.

[6]MEDMCQA from the All India Institute of Medical Sciences, and MEDQA from the U.S. National Medical Board Examination.

Table 1: Image classification results. We report the average performance across 10 experiments, with PGR calculated as described in Appendix D.6. *Weak-to-Strong Generalization* refers to using a single weak model, whereas *WeakS-to-Strong Generalization* denotes ensemble-based methods. We highlight the best score in red and the second-best score in **bold**. Model(3) refers to the use of three weak models. For a fair comparison with Co-sup, which utilizes 7 weak models for ɪWɪʟᴅCᴀᴍ, ꜰMᴏW and ɪMᴀɢᴇNᴇᴛ, and 5 for CᴀᴍᴇʟʏᴏN17, we report the RAVEN(>3) performance achieved using the same number of weak models as Co-sup. * indicates that the implementation code was created by us. The standard deviations are reported in Appendix G.1.

| | | Weak model | Weak-to-Strong Generalization | | | | WeakS-to-Strong Generalization | | | | | Strong model |
|---|---|---|---|---|---|---|---|---|---|---|---|---|
| | | AlexNet | Naive | Conf | Boots | V-sup | Ens(3)* | Bayes(3)* | RAVEN(3) | Co-sup(>3) | RAVEN(>3) | DINO ViT-B/8 |
| *Robust Weak-to-Strong Generalization (Out-of-distribution)* | | | | | | | | | | | | |
| ɪWɪʟᴅCᴀᴍ | Accuracy | 43.83 | 46.82 | 47.97 | 48.76 | 47.92 | 49.38 | 49.96 | **52.79** | 49.83 | 55.46 | 94.76 |
| | PGR | - | 4.21 | 6.52 | 8.08 | 6.40 | 9.73 | 10.90 | **16.25** | 10.03 | 22.14 | - |
| CᴀᴍᴇʟʏᴏN17 | Accuracy | 66.90 | 67.83 | 68.90 | 70.84 | 67.85 | 72.18 | 68.74 | 73.67 | 70.00 | 73.45 | 97.93 |
| | PGR | - | 2.47 | 5.90 | 12.24 | 2.51 | 17.13 | 5.13 | 21.46 | 14.13 | 21.48 | - |
| ꜰMᴏW | Accuracy | 26.18 | 25.68 | 24.75 | 25.97 | 25.57 | 27.54 | 25.89 | **29.46** | 28.70 | 31.85 | 59.72 |
| | PGR | - | -1.26 | -4.02 | -0.38 | -1.60 | 4.31 | -0.67 | 10.06 | 13.24 | 14.06 | - |
| **Avg.** | Accuracy | 45.64 | 46.78 | 47.21 | 48.53 | 47.11 | 49.70 | 47.78 | 51.54 | 49.51 | 53.23 | 81.20 |
| | PGR | - | 1.80 | 2.80 | 6.65 | 2.44 | 10.39 | 2.77 | 16.09 | 12.47 | 19.27 | - |
| *Weak-to-Strong Generalization (In-distribution)* | | | | | | | | | | | | |
| ɪMᴀɢᴇNᴇᴛ | Accuracy | 54.90 | 66.83 | 68.26 | 65.72 | 67.88 | 67.11 | 49.98 | 67.90 | 68.60 | 68.35 | 74.49 |
| | PGR | - | 60.97 | 68.12 | 55.39 | 65.87 | 62.02 | -23.99 | 66.33 | 69.54 | 68.60 | - |

in InD W2S performance. Generally, ensemble-based methods outperform those relying on a single weak model; nevertheless, RAVEN is far more effective than other ensemble-based approaches, achieving 55% average improvement in PGR and 3.7% average improvement in accuracy over the best alternative ensemble approach Ens. In contrast to other baselines that do not use domain information, Co-sup utilizes domain information by training each weak model on a specific group of domains. Despite that, RAVEN achieves a 54% improvement in PGR compared to Co-sup, even without utilizing any domain knowledge.

**Text classification.** Experiments on the text classification tasks (Table 2) show that RAVEN consistently outperforms all baselines. In the OOD setting, RAVEN achieves a 57% average improvement in PGR and a 1.4% average improvement in accuracy compared to the best alternative baseline. In the InD setting, RAVEN yields a 27% average improvement in PGR over the best baseline. We further observe that Bayes [19], despite leveraging multiple weak models, underperforms in this context and can even fall below the performance of individual weak models. Note that the OOD setting involves more fine-tuning instances than the InD setting, which contributes to the higher PGR observed for OOD tasks.

Table 2: Text classification results. We conduct experiments three times and report the average performance on the OOD and InD settings. We highlight the best score in red and the second-best score in **bold**. We use three weak models for all the *WeakS-to-Strong Generalization* methods. The standard deviations are reported in Section G.2.

| | | Weak model | Weak-to-Strong Generalization | | | | WeakS-to-Strong Generalization | | | | Strong model |
|---|---|---|---|---|---|---|---|---|---|---|---|
| | | | Naive | Conf | Boots | V-sup | Ens(3)* | Bayes(3)* | Co-sup(3) | RAVEN(3) | |
| AMᴀᴢᴏN-Wɪʟᴅs *(Out-of-distribution)* | | | | | | | | | | | |
| Llama-3.2-1B → Llama-3.1-8B | Accuracy | 68.17 | 68.60 | 68.19 | 68.59 | 67.88 | 68.45 | 67.03 | **69.95** | 71.14 | 72.15 |
| | PGR | – | 10.88 | 0.50 | 10.63 | -7.36 | 6.95 | -28.70 | **44.77** | 74.56 | – |
| Qwen2.5-0.5B → Qwen2.5-7B | Accuracy | 66.34 | 67.69 | 67.12 | **67.79** | 67.26 | 67.64 | 64.44 | 67.60 | 70.15 | 70.78 |
| | PGR | – | 30.35 | 17.58 | **32.61** | 20.74 | 29.15 | -43.01 | 28.32 | 85.88 | – |
| Qwen2.5-0.5B → Qwen2.5-14B | Accuracy | 66.34 | 70.97 | 70.82 | 70.00 | 70.20 | 70.52 | 59.07 | **70.98** | 71.34 | 75.66 |
| | PGR | – | 49.70 | 48.00 | 39.20 | 41.30 | 44.80 | -78.00 | **49.80** | 53.60 | – |
| MᴇᴅMCQA → MᴇᴅQA *(Out-of-distribution)* | | | | | | | | | | | |
| Qwen2.5-0.5B → Meditron-7B | Accuracy | 25.70 | 26.00 | **26.30** | 25.80 | 25.70 | 25.70 | 24.70 | 25.30 | 26.50 | 27.60 |
| | PGR | – | 17.90 | **30.50** | 5.30 | 1.60 | -2.60 | -52.60 | -23.20 | 42.60 | – |
| Qwen2.5-0.5B → Meditron-70B | Accuracy | 25.70 | **27.57** | 27.54 | 27.32 | 27.26 | 26.08 | 20.90 | 25.22 | 27.73 | 36.37 |
| | PGR | – | **17.50** | 17.20 | 15.20 | 14.60 | 3.60 | -45.00 | -4.50 | 19.00 | – |
| AMᴀᴢᴏN-Wɪʟᴅs *(In-distribution)* | | | | | | | | | | | |
| Llama-3.2-1B → Llama-3.1-8B | Accuracy | 69.74 | 70.23 | 70.02 | 70.06 | 70.02 | **70.33** | 66.29 | 69.97 | 70.44 | 71.33 |
| | PGR | – | 30.95 | 17.26 | 20.00 | 17.68 | **37.26** | -217.89 | 14.32 | 44.00 | – |
| Qwen2.5-0.5B → Qwen2.5-7B | Accuracy | 67.71 | **68.61** | 68.18 | 68.32 | 68.10 | 68.50 | 63.23 | 68.56 | 68.94 | 69.42 |
| | PGR | – | **52.59** | 27.37 | 35.97 | 23.07 | 46.14 | -262.76 | 50.05 | 71.95 | – |

**Preference alignment.** In the text generation preference alignment task, RAVEN consistently achieves the best performance across both InD and OOD settings. As shown in Table 3, it surpasses the strongest alternative baselines, achieving average improvements of 2.7% and 32.8% in the OOD settings, and 1.2% and 25.6% in the InD setting (WR and PGR, respectively). Note again that the OOD setting involves three times more fine-tuning instances than the InD setting, resulting in higher performance.

Table 3: Preference alignment results.

| | Weak | Naive | Ens(3)* | Bayes(3)* | RAVEN(3) | Strong |
|---|---|---|---|---|---|---|
| HELPFULNESS → HARMLESSNESS (*Out-of-distribution*) | | | | | | |
| WR | 59.82 | 61.38 | **62.73** | 61.41 | **64.04** | 66.98 |
| PGR | - | 21.79 | **40.64** | 22.21 | **58.94** | - |
| SUMMARIZATION → HUMAN-LIKE (*Out-of-distribution*) | | | | | | |
| WR | 57.50 | 67.86 | 68.09 | **68.18** | **70.37** | 83.00 |
| PGR | - | 40.63 | 41.53 | **41.88** | **50.47** | - |
| HARMLESSNESS → HARMLESSNESS (*In-distribution*) | | | | | | |
| WR | 59.82 | 61.49 | 61.43 | **62.83** | **63.60** | 66.98 |
| PGR | - | 23.32 | 22.49 | **42.04** | **52.79** | - |

## 5.3 Further analysis

We conduct additional analyses of RAVEN on the image classification task within the robust W2S scenario. Following the setup in Table 1, we conduct ten experiments and report their average unless stated otherwise.

**Ablation study.** We conduct ablation studies to evaluate the impact of ensembling, easy-sample-guided initialization, and adaptive weighting introduced in RAVEN. We incrementally incorporate each component and evaluate performance. As shown in Table 4, all components effectively contribute to RAVEN's performance, validating our design choices and their individual importance.

Table 4: Ablation study. The values represent the averages across all datasets. Detailed per-dataset results are available in Appendix H.6.

| Ensemble | | ✓ | ✓ | ✓ | ✓ |
|---|---|---|---|---|---|
| Easy-sample guided init. | | | ✓ | | ✓ |
| Adaptive weighting | | | | ✓ | ✓ |
| Accuracy | 46.78 | 49.76 | 50.51 | 51.37 | **51.97** |
| PGR | 1.80 | 10.39 | 10.67 | 14.28 | **16.09** |

**Training scheduling.** RAVEN first trains only the $\Theta_S$ parameters on easy samples using $\mathcal{L}_{\text{ensemble}}$, and then updates both $\Theta_S$ and the ensembling weights $\Theta_W$ on the entire dataset using $\mathcal{L}_{\text{adaptation}}$. We further explore how different strategies for using easy samples and for applying static vs. adaptive weighting affect performance. As shown in Table 5, our strategy of easy-sample guided initialization with adaptive weighting achieves the best performance, confirming the effectiveness of our approach. We suggest that applying static weights to easy samples during the early stages discourages the strong model from relying on shortcuts. These shortcuts occur when the strong model learns a weak signal combination that simply mimics its own initial suboptimal predictions, an easy way to minimize the cross-entropy loss without genuine learning. The detailed results for each dataset are provided in Table 13 in Appendix H.1.

Table 5: Sample and weight scheduling. *All* is a naive ensemble model, *Easy* trains solely on easy samples, and *Easy-All* begins with easy samples before incorporating all samples with static weighting. *Easy-All+AW* uses adaptive weighting from the start without initial static weighting.

| Metric | All | Easy | Easy-All | Easy-All+AW | RAVEN |
|---|---|---|---|---|---|
| Accuracy | 49.76 | 50.10 | 50.51 | 50.86 | **51.97** |
| PGR | 10.39 | 11.13 | 10.67 | 12.85 | **16.09** |

**Increased weak model diversity.** To enhance the diversity of weak models, we conduct additional experiments using *(i)* different weak model architectures and *(ii)* different data sources for training the weak models. For different architectures, we use AlexNet, ResNet18, and SqueezeNet, as weak models to supervise the strong model DINO ViT-B/8. Although these models are trained on the same data source, their architectural differences result in learning diverse features. For different data sources, we follow the setup in Liu and Alahi [45], which constructs distinct data splits based on sub-domains. Specifically, we train seven AlexNet models on different "camera location" domains for IWILDCAM, five AlexNet models on different "hospital" domains for CAMELYON17, and seven AlexNet models on different "time" domains for FMOW. As shown in Table 6, RAVEN achieves an 81% PGR improvement in the setting with different architectures and a 24% PGR improvement in

Table 6: Performance comparison across different weak model configurations. Reported values are averaged over all datasets, with detailed results for each dataset provided in Appendix H.2.

| | *Weak model* | *Weak-to-Strong Generalization* | | | | *WeakS-to-Strong Generalization* | | | | *Strong model* |
|---|---|---|---|---|---|---|---|---|---|---|
| | | Naive | Conf | Boots | V-sup | Ens | Bayes | Co-sup | RAVEN | DINO ViT-B/8 |
| | | *Different Architectures of Weak Models* | | | | | | | | |
| Accuracy | 42.04 | 42.80 | 41.18 | 44.21 | 41.87 | 43.40 | **44.23** | 42.45 | **46.03** | 84.43 |
| PGR | - | 1.78 | -1.72 | 3.15 | -0.26 | 3.28 | **5.33** | -0.31 | **9.65** | - |
| | | *Different Data Sources for Training Weak Models* | | | | | | | | |
| Accuracy | 47.40 | 46.34 | 46.92 | 47.33 | 46.46 | 49.40 | **50.13** | 47.43 | **51.17** | 84.63 |
| PGR | - | -2.08 | -0.50 | 0.93 | -1.75 | 5.46 | **7.76** | 1.05 | **10.49** | - |

the setting with different data sources, compared to the best alternative baseline Bayes. These results confirm that RAVEN is effective at weak-to-strong generalization with weak models of both low and substantial diversity.

**How do the weights $\Theta_W$ evolve over iterations?** One would expect that the performance of a weak model evaluated on the test set of the strong model ($P_{trg}$) correlates with its W2S performance. Consequently, it would be desirable for RAVEN to assign higher weights to weak models with better $P_{trg}$ performance. To investigate this, we begin by examining the correlation between the performance of weak models on $P_{trg}$ and their W2S performance and indeed observe high correlation, highlighting the importance of selecting the optimal weak model (Figure 4 *Left*).

We next aim to understand whether the weights $\Theta_W$ that the strong model assigns to weak models in RAVEN agree well with the actual $P_{trg}$ performance of the weak models. Notably, we find that the strong model predominantly assigns the highest weight to the best-performing weak model for $P_{trg}$, despite not having access to information about the weak models' performance as evaluated against GT (Figure 4 *Right*). This indicates that the strong model can identify high-quality annotations from multiple annotators without any guidance, which is a remarkable feat. Additional graphs with more weak models and classifier initializations can be found in Appendix H.4 and Appendix H.5.

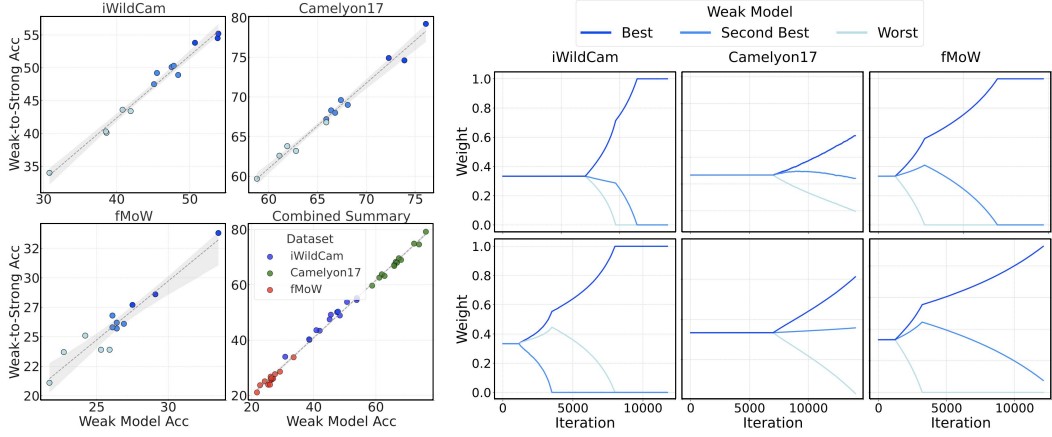

Figure 4: *Left*: Correlation between weak model accuracy $Acc(P_{trg}; f_w^{\text{src}})$ and its W2S accuracy $Acc(P_{trg}; f_s^{\text{pseudo}})$. *Right*: The weights $\Theta_W$ assigned to weak models across two runs of RAVEN.

**Number of weak models $M$.** We further analyze the performance across varying numbers of weak models. As the number of weak models increases, we observe that the performance of RAVEN improves (Figure 5). However, the gains gradually diminish after including a large number of weak models, indicating a saturation effect.

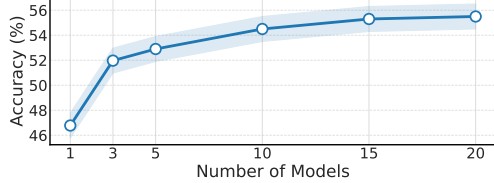

Figure 5: Effect of the number of weak models.

**Other results.** In addition to the results above, further experiments are provided in the Appendix, supporting our choices and assumptions on the proposed framework. Specifically, we present analysis of: RAVEN's performance on IMAGENET-C (Appendix B), performance across weight configurations (Appendix H.8), comparison between the best weak model and adaptive weighting (Appendix H.7), exploration of strong model's identification of the best weak model (Appendix H.10), scaling analysis using diverse (weak, strong) model pairs (Appendix H.11), and qualitative preference-alignment results (Appendix H.12).

## 6 Related work

**Weak-to-strong (W2S) generalization.** W2S generalization framework was first introduced in [10]. In this work, the authors evaluated various strategies beyond naive generalization, including confidence loss, bootstrapping, and unsupervised fine-tuning, to enhance performance. They demonstrated the feasibility of developing superhuman models, a finding that was further supported by subsequent studies [67, 64]. Theoretical insights behind this observation were investigated in [13, 38, 28, 60, 23, 66]. To enhance W2S generalization performance, [26] extended the concept of confidence loss introduced in [10], developing adaptive confidence loss, which dynamically adjusts the balance between learning from weak model confidence. Multiple weak models have been employed using the AdaBoost technique [1], within the framework of hierarchical mixtures of experts [45], and in estimating Bayesian-based confidence loss [19]. Sang et al. [57] leveraged debates among multiple models to enhance strong models. Within this ensemble framework, we propose a method for identifying reliable weak supervision, which proves especially effective in our novel scenario involving distribution shifts.

**Scalable oversight.** Scalable oversight [3, 6, 42] aims to enhance human supervision in novel and challenging environments. To this end, models can be used to evaluate other models [29, 58, 33, 32] or to decompose complex problems into simpler subproblems [41, 44]. By contrast, W2S generalization explores how to effectively leverage unreliable human supervision itself to train models that can ultimately surpass human performance.

**Learning under distribution shift.** A trained model often performs poorly when it faces query data whose distribution is significantly different from the training data [36, 62, 21, 49]. Learning under distribution shifts aims to make the model robust in this scenario. This challenge has been observed across domains such as healthcare, autonomous driving, and facial recognition, where models fail to generalize across hospitals, lighting conditions, or demographic subgroups due to distribution mismatches [35, 53, 2, 20, 9]. A range of methods has been proposed to address this issue, and their effectiveness is actively evaluated through diverse benchmarks [25, 63, 30].

In the context of existing literature, our novel scenario can be viewed as an intersection of **weak-to-strong generalization** and **learning under distribution shift**.

## 7 Concluding remarks

*Limitations.* The weak models used in RAVEN differ only in their random seeds. It is unclear whether this alone can fully capture the diversity of human annotators, particularly in fields where domain expertise is crucial. Another aspect that calls for further investigation is the choice of adaptive weights (Appendix H.8). While we found that model-wise weights outperform the (model, sample)-wise variant, the former is inherently a subset of the latter (where model-wise weights are the same across all samples). Future work could explore ways to address the optimization challenges of (model, sample)-wise weighting, potentially leading to an even stronger RAVEN variant.

*Conclusion.* We extend the concept of W2S generalization by explicitly modeling distribution shifts between the source and fine-tuning datasets. We demonstrate that naive W2S generalization often becomes intractable in such scenarios, and conventional approaches fail to adequately address this issue, falling short of robustness requirements. To address this challenge, we present RAVEN, a novel framework that enables robust W2S generalization.

**Acknowledgments.** We would like to thank Artyom Gadetsky, Fabian Gröger, Maxim Kodryan, Ramon Vinas Torné, Shuo Wen, Siba Smarak Panigrahi, and Yulun Jiang for their valuable discussions

regarding our work. We gratefully acknowledge the support of the Swiss National Science Foundation (SNSF) starting grant TMSGI2_226252/1, SNSF grant IC00I0_231922, the Swiss AI Initiative and the CIFAR Multiscale Human Catalyst. Myeongho Jeon was supported by the National Research Foundation of Korea (NRF) grant funded by the Korea government (MSIT) [RS-2024-00337693, N10250156]. Suhwan Choi was supported by 1) the Starting growth Technological R&D Program (RS-2024-00506994) funded by the Ministry of SMEs and Startups (MSS, Korea), 2) Culture, Sports and Tourism R&D Program through the Korea Creative Content Agency grant funded by the Ministry of Culture, Sports and Tourism in 2024 (Project Name: Development of K-POP artist-centered video editing solution: customized multimodal AI model and generative asset, Project Number: RS-2024-00399433, Contribution Rate: 50%), 3) Artificial intelligence industrial convergence cluster development project funded by the Ministry of Science and ICT (MSIT, Korea) & Gwangju Metropolitan City, and 4) Institute of Information & communications Technology Planning & Evaluation (IITP) grant funded by the Korea government (MSIT) [NO.RS-2021-II211343, Artificial Intelligence Graduate School Program (Seoul National University)]. Jan Sobotka was supported by the Bakala Foundation during his studies at EPFL.

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

# A    Definition of distribution shift

To define distribution shift between some two distributions $P_{\text{src}}$ and $P_{\text{tuning}}$, we use the generalization gap: $\Delta R(f) = R_{\text{tuning}}(f) - R_{\text{src}}(f)$, where $R(f) = \mathbb{E}_{(x,y) \sim P}[\mathcal{L}(f(x), y)]$ is the expected risk, $f$ denotes the classifier, and $\mathcal{L}$ is the loss function (*e.g.*, cross-entropy). A large $\Delta R(f)$ indicates that the model struggles to generalize from distribution $P_{\text{src}}$ to distribution $P_{\text{tuning}}$, showing the presence of a significant distribution shift.

Another way to quantify distribution shift is through a divergence metric $D(P_{\text{src}}, P_{\text{tuning}})$, where $D$ can be any suitable measure such as Maximum Mean Discrepancy (following [54]), KL-divergence, or optimal transport. When $D(P_{\text{src}}, P_{\text{tuning}}) \leq \epsilon$ and $\epsilon \in \mathbb{R}_+$ is small enough, we regard the distributions as approximately aligned, *i.e.*, $P_{\text{src}} \approx P_{\text{tuning}}$. Conversely, we say that $P_{\text{src}} \not\approx P_{\text{tuning}}$ and consider the shift to be significant—potentially misleading the model's performance at inference—when $D(P_{\text{src}}, P_{\text{tuning}}) > \epsilon$.

# B    Motivating experiment

**Without fine-tuning.** For image classification tasks, Burns et al. [10] used an IMAGENET-pretrained AlexNet as the weak model without additional training. To replicate their setup, we compute the mean and standard deviation of the pre-trained AlexNet's performance over five random splits of the IMAGENET validation set into $P_{tuning}$ and $P_{trg}$, and over five random splits of the IMAGENET-C dataset into $P_{tuning}$ and $P_{trg}$. The results of this experiment are provided in Table 7 and, combined with the setup described below, shown in Figure 2.

**With fine-tuning.** A more common practice, however, is to evaluate models trained with different random seeds while keeping the dataset fixed. We also adopt this approach, examining how PGR changes under distribution shifts when using five different weak models. Specifically, using different random seeds, we reinitialize the classification head of the IMAGENET-pretrained AlexNet and train it for 20 epochs. This approach is particularly relevant in the weakS-to-strong generalization framework (ensemble-based methods), where multiple distinct weak models are required. Using these fine-tuned models, we compare RAVEN to naive W2S generalization on the IMAGENET-C dataset. As shown in Table 8, despite severe synthetic domain shifts, the performance of RAVEN exceeds that of the weak model (human in the future) in most cases, outperforming the naive W2S generalization method. For this experiment, we use the same AlexNet weak models as reported in Table 1, and designate DINO ResNet50 as the strong model.

Finally, the results for the naive W2S generalization method shown in Figure 2 are the average over these two setups, effectively balancing clarity of presentation, adherence to previous work [10], and common practices used in the field.

Table 7: Result of Motivating Experiment: without fine-tuning of weak models. For the weak and strong models, we report the accuracy, while for the W2S fine-tuning (AlexNet → DINO ResNet50), we present the PGR. The values represent the mean and standard deviation over five runs.

| Corruption Type | AlexNet | DINO ResNet50 | AlexNet → DINO ResNet50 (PGR) |
|---|---|---|---|
| No-shift | $56.26 \pm 0.56$ | $64.00 \pm 0.39$ | $74.01 \pm 5.67$ |
| Saturate | $45.72 \pm 0.43$ | $59.68 \pm 0.43$ | $44.43 \pm 1.83$ |
| Brightness | $42.70 \pm 0.37$ | $58.62 \pm 0.69$ | $39.59 \pm 1.02$ |
| Elastic | $40.18 \pm 0.28$ | $52.64 \pm 0.36$ | $19.56 \pm 2.34$ |
| JPEG | $37.66 \pm 0.25$ | $53.00 \pm 0.79$ | $9.63 \pm 2.38$ |
| Gaussian Blur | $13.64 \pm 0.17$ | $41.24 \pm 0.56$ | $5.87 \pm 1.18$ |
| Pixelate | $29.06 \pm 0.42$ | $49.44 \pm 0.47$ | $5.50 \pm 0.66$ |
| Defocus Blur | $11.54 \pm 0.15$ | $39.32 \pm 0.71$ | $5.48 \pm 0.80$ |
| Spatter | $23.08 \pm 0.30$ | $48.30 \pm 0.30$ | $3.00 \pm 0.76$ |
| Contrast | $9.18 \pm 0.25$ | $52.04 \pm 0.41$ | $2.66 \pm 0.63$ |
| Fog | $13.86 \pm 0.36$ | $47.34 \pm 0.34$ | $1.91 \pm 0.28$ |
| Motion Blur | $16.00 \pm 0.39$ | $38.76 \pm 0.19$ | $1.57 \pm 1.92$ |
| Snow | $12.52 \pm 0.25$ | $39.54 \pm 0.44$ | $1.56 \pm 1.37$ |
| Zoom Blur | $19.22 \pm 0.65$ | $39.94 \pm 0.45$ | $1.04 \pm 1.79$ |
| Speckle Noise | $8.58 \pm 0.08$ | $41.94 \pm 0.27$ | $-0.24 \pm 0.54$ |
| Gaussian Noise | $5.06 \pm 0.17$ | $40.06 \pm 0.37$ | $-1.03 \pm 0.67$ |
| Shot Noise | $5.22 \pm 0.16$ | $39.64 \pm 0.38$ | $-1.45 \pm 0.46$ |
| Frost | $10.68 \pm 0.13$ | $36.90 \pm 0.37$ | $-1.69 \pm 1.11$ |
| Impulse Noise | $4.18 \pm 0.19$ | $37.12 \pm 0.27$ | $-1.76 \pm 0.58$ |
| Glass Blur | $10.68 \pm 0.13$ | $30.92 \pm 0.41$ | $-9.00 \pm 0.80$ |

Table 8: Performance comparison on IMAGENET-C for the Motivating Experiment: with fine-tuning of weak models. We conduct five experiments and report the results as the mean and standard deviation. Negative PGRs are highlighted in red.

| Corruption | Metric | AlexNet | Naive | RAVEN | Dino ResNet50 |
|---|---|---|---|---|---|
| **Saturate** | Accuracy | $44.34 \pm 0.44$ | $47.66 \pm 0.51$ | $49.23 \pm 0.40$ | $60.06 \pm 0.39$ |
| | PGR | - | $20.94 \pm 1.09$ | $31.22 \pm 1.42$ | - |
| **Brightness** | Accuracy | $40.64 \pm 0.36$ | $43.72 \pm 0.50$ | $45.67 \pm 0.06$ | $58.90 \pm 0.50$ |
| | PGR | - | $16.93 \pm 0.74$ | $27.63 \pm 0.62$ | - |
| **Elastic** | Accuracy | $38.02 \pm 0.49$ | $37.82 \pm 0.38$ | $38.83 \pm 0.06$ | $52.90 \pm 0.70$ |
| | PGR | - | $-1.34 \pm 0.78$ | $5.07 \pm 2.07$ | - |
| **JPEG** | Accuracy | $35.74 \pm 0.48$ | $35.02 \pm 0.35$ | $36.27 \pm 0.06$ | $53.16 \pm 0.79$ |
| | PGR | - | $-4.42 \pm 0.36$ | $2.94 \pm 0.96$ | - |
| **Gaussian blur** | Accuracy | $10.86 \pm 0.32$ | $10.92 \pm 0.42$ | $11.57 \pm 0.35$ | $41.84 \pm 0.35$ |
| | PGR | - | $0.22 \pm 0.57$ | $2.40 \pm 0.88$ | - |
| **Pixelate** | Accuracy | $26.46 \pm 0.43$ | $26.12 \pm 0.70$ | $27.47 \pm 0.47$ | $49.72 \pm 0.46$ |
| | PGR | - | $-1.48 \pm 0.55$ | $4.54 \pm 1.38$ | - |
| **Defocus blur** | Accuracy | $9.22 \pm 0.27$ | $9.32 \pm 0.34$ | $9.67 \pm 0.49$ | $40.16 \pm 0.50$ |
| | PGR | - | $0.57 \pm 0.34$ | $1.62 \pm 1.30$ | - |
| **Spatter** | Accuracy | $22.74 \pm 0.45$ | $21.90 \pm 0.47$ | $22.97 \pm 0.38$ | $48.90 \pm 0.46$ |
| | PGR | - | $-3.18 \pm 0.26$ | $0.72 \pm 1.14$ | - |
| **Contrast** | Accuracy | $8.40 \pm 0.19$ | $8.46 \pm 0.21$ | $8.97 \pm 0.06$ | $52.40 \pm 0.47$ |
| | PGR | - | $0.20 \pm 0.12$ | $1.34 \pm 0.24$ | - |
| **Fog** | Accuracy | $13.92 \pm 0.48$ | $14.04 \pm 0.55$ | $14.17 \pm 0.06$ | $47.64 \pm 0.62$ |
| | PGR | - | $-2.75 \pm 0.34$ | $0.33 \pm 0.06$ | - |
| **Motion blur** | Accuracy | $13.04 \pm 0.21$ | $12.06 \pm 0.21$ | $13.10 \pm 0.10$ | $39.18 \pm 0.48$ |
| | PGR | - | $-3.78 \pm 0.30$ | $0.34 \pm 0.60$ | - |
| **Snow** | Accuracy | $12.08 \pm 0.22$ | $11.58 \pm 0.16$ | $12.27 \pm 0.29$ | $40.34 \pm 0.38$ |
| | PGR | - | $-1.93 \pm 0.26$ | $0.55 \pm 0.95$ | - |
| **Zoom blur** | Accuracy | $16.02 \pm 0.35$ | $14.66 \pm 0.28$ | $16.30 \pm 0.39$ | $40.64 \pm 0.46$ |
| | PGR | - | $-5.36 \pm 0.42$ | $1.30 \pm 1.56$ | - |
| **Speckle noise** | Accuracy | $8.12 \pm 0.36$ | $7.70 \pm 0.37$ | $8.20 \pm 0.17$ | $42.88 \pm 0.36$ |
| | PGR | - | $-1.15 \pm 0.25$ | $0.10 \pm 0.25$ | - |
| **Gaussian noise** | Accuracy | $4.68 \pm 0.26$ | $3.90 \pm 0.23$ | $4.90 \pm 0.10$ | $40.10 \pm 0.00$ |
| | PGR | - | $-2.23 \pm 0.15$ | $0.38 \pm 0.16$ | - |
| **Shot noise** | Accuracy | $4.76 \pm 0.26$ | $4.12 \pm 0.16$ | $5.03 \pm 0.06$ | $40.10 \pm 0.00$ |
| | PGR | - | $-1.95 \pm 0.14$ | $0.54 \pm 0.14$ | - |
| **Frost** | Accuracy | $10.50 \pm 0.41$ | $9.28 \pm 0.66$ | $9.90 \pm 0.35$ | $36.88 \pm 0.42$ |
| | PGR | - | $-4.27 \pm 0.42$ | $-1.36 \pm 0.21$ | - |
| **Impulse noise** | Accuracy | $4.00 \pm 0.20$ | $3.40 \pm 0.07$ | $4.23 \pm 0.06$ | $38.28 \pm 0.38$ |
| | PGR | - | $-1.95 \pm 0.10$ | $0.49 \pm 0.34$ | - |
| **Glass blur** | Accuracy | $9.56 \pm 0.36$ | $7.82 \pm 0.33$ | $8.27 \pm 0.25$ | $31.80 \pm 0.35$ |
| | PGR | - | $-7.78 \pm 0.17$ | $-5.79 \pm 0.61$ | - |

## C    Theoretical analysis of how RAVEN operates

To understand why RAVEN often identifies the most suitable weak model for the target data distribution, we begin with the following assumption.

**Assumption 1.** The ensemble of weak models tends to make predictions similar to those of the best-performing weak model:

$$\arg \max_{k \in \{1,\ldots,C\}} f_{\Sigma}(x)[k] \approx \arg \max_k f^*(x)[k], \tag{5}$$

where

$$f_{\Sigma}(x) := \sum_{i=1}^{M} f_{w_i}^{\text{src}}(x),$$

and $f^*(x)$ denotes the best-performing weak model. While the ensemble prediction is often close to that of the best model, we assume that $f^*$ is more accurate and confident.

Let $f_s$ denote the strong model, and define the adaptation loss as:

$$L_{\text{adaptation}}(\Theta_S, \Theta_W) := L_{\text{CE}} \left( f_s(x), \sum_{i=1}^{M} \theta^{(i)} f_{w_i}^{\text{src}}(x) \right) \quad \text{s.t.} \quad \sum_{i=1}^{M} \theta^{(i)} = 1, \ \theta^{(i)} \in \Theta_W. \tag{6}$$

This loss is optimized by alternating the following two steps:

**Step 1.** *Optimize $\Theta_S$ with $\Theta_W$ fixed.*

When the weights are uniform (*i.e.*, $\theta^{(i)} = \frac{1}{M}$), the strong model learns to mimic the average behavior of weak models:

$$f_s(x) \approx \frac{1}{M} \sum_{i=1}^{M} f_{w_i}^{\text{src}}(x). \tag{7}$$

**Step 2.** *Optimize $\Theta_W$ with $\Theta_S$ fixed.*

We can rewrite the objective as:

$$L_{\text{CE}}(\Theta_W) = \sum_{i=1}^{M} \theta^{(i)} C_i, \tag{8}$$

where

$$C_i := -\mathbb{E}_{x \sim D} \left[ \sum_{k=1}^{C} f_{w_i}^{\text{src}}(x)[k] \log f_s(x)[k] \right]. \tag{9}$$

**Assumption 2.** Each weak model is highly confident in predicting a single class:

$$f_{w_i}^{\text{src}}(x) = (\epsilon, \ldots, 1 - \bar{\epsilon}, \ldots, \epsilon), \quad \epsilon \ll 1, \tag{10}$$

with the largest value $1 - \bar{\epsilon}$ at index

$$\hat{k}_i := \arg \max_k f_{w_i}^{\text{src}}(x)[k].$$

Then,

$$\sum_{k=1}^{C} f_{w_i}^{\text{src}}(x)[k] \log f_s(x)[k] \approx \log f_s(x)[\hat{k}_i], \tag{11}$$

and the objective simplifies to:

$$C_i \approx -\mathbb{E}_{x \sim D} \left[ \log f_s(x)[\hat{k}_i] \right]. \tag{12}$$

By Assumption 1 and Eq. (3), we infer that the best weak model $f^*(x)$ most closely matches the predictions of $f_s(x)$ and therefore receives the highest weight—consistent with empirical findings. Notably, this conclusion still holds even when only the best weak model satisfies Assumption 2, since the best weak model yields the smallest $C_i$. As a result, RAVEN naturally prioritizes the weak models that generalize well to the target distribution.

Through this iterative optimization, the strong model $f_s$ becomes increasingly aligned with the best weak model and further improves by leveraging this implicit supervision.

# D  Experimental details

## D.1  Adaptive weighting

We initialize the adaptive weights $\theta^{(i)} \in \Theta_W$ uniformly, $i.e.$, $\theta^{(i)} = \frac{1}{M}$ for $i \in \{1, \ldots M\}$. After the easy-sample guided initialization phase, during which $\theta^{(i)}$ are fixed and only the strong model parameters $\Theta_S$ are trained, we start optimizing the adaptive weights as well. More specifically, for each minibatch of the fine-tuning data, we first generate and combine soft labels from the $M$ weak models using the latest $\Theta_W$ (Eq. 3). Then, using these pseudo labels, we calculate gradients of $\mathcal{L}_{\text{adaptation}}(\Theta_S, \Theta_W)$ with respect to $\Theta_S$ and $\Theta_W$, and perform a single update step of both using the Adam optimizer [34] and SGD, respectively. After each update of $\Theta_W$, we clip and normalize the adaptive weights to sum up to 1 as:

$$\theta_{\text{new}}^{(i)} = \frac{\max(\theta^{(i)}, \epsilon)}{\sum_{j=1}^{M} \max(\theta^{(j)}, \epsilon)}, \tag{13}$$

where $\epsilon = 10^{-6}$. $\theta_{\text{new}}^{(i)}$ are then set as $\theta^{(i)}$ for the next minibatch of the fine-tuning process.

For text classification, we derive the weak supervision $y_{\text{soft}}(x)$ ($i.e.$, pseudo-label) by computing a weighted sum of weak model logits $y_{\text{aw}}$ using adaptive weights, followed by a softmax operation for stabilization:

$$y_{\text{soft}}^{(c)}(x) = \frac{\exp\left(y_{\text{aw}}^{(c)}(x)\right)}{\sum_{k=1}^{K} \exp\left(y_{\text{aw}}^{(k)}(x)\right)}, \tag{14}$$

$$\text{where} \quad y_{\text{aw}}(x) = \sum_{i=1}^{M} \theta^{(i)} f_{w_i}^{src}(x). \tag{15}$$

Above, $f_{w_i}^{src}(x)$ represents the logits of the $i$-th weak model trained on source data, $K$ is the number of label classes, and $c$ denotes a specific class label. Consequently, $y_{\text{soft}}(x)$ serves as the weak supervision.

All hyperparameters, including those of the baselines, are found using a grid search based on the validation loss. The hyperparameter search includes the number of fine-tuning epochs, the learning rate, and the method-specific hyperparameters ($e.g.$, the learning rate of the adaptive weights and the length of the easy-sample guided initialization phase). Building on the publicly available code from Burns et al. [10], we use the cosine annealing schedule [46] while optimizing the strong model parameters $\Theta_S$, but keep the learning rate of the adaptive weights $\Theta_W$ fixed. We perform a hyperparameter search in the range shown in Table 9.

Table 9: Hyperparameter configurations for RAVEN.

| Task | Learning rate for $\Theta_S$ | Learning rate for $\Theta_W$ | Easy-sample guided period | Epoch |
|------|------|------|------|------|
| Image Classification | {1e-6, 1e-5, 1e-4, 1e-3} | {1e-6, 1e-5, 1e-4, 1e-3} | {10%, 20%, 50%} | {30, 50, 80, 100} |
| Text Classification | {1e-6, 1e-5, 1e-4} | {1e-5, 1e-4, 1e-3} | {20%, 50%} | {10, 20} |
| Preference Alignment | {1e-6, 5e-6, 1e-5, 5e-5} | {1e-5, 1e-4, 1e-3} | {20%, 50%} | {2} |

## D.2  Weak models for ensemble-based methods

The weak models used in ensemble-based methods are trained using different seeds and hyperparameters found through a search over the number of epochs, learning rate, and weight decay. For image classification experiments, we train weak models with the Adam optimizer [34], the cross-entropy loss function, early stopping, and multiplicative (factor 0.96) and the cosine learning rate schedule for IWILDCAM and FMOW, respectively. For CAMELYON17 and IMAGENET, we employ SGD with a momentum of 0.9 and a fixed learning rate. Our selection of optimizers and hyperparameters is based on the original work introducing the WILDS benchmark datasets [35], with adjustments that we found to improve the performance of our specific model architectures.

With this setup, all weak models achieve around the same InD validation accuracy on their respective datasets ($\pm 2\%$). Since the W2S fine-tuning step of the IMAGENET experiments employs its validation dataset, we use 10% of the IMAGENET training set as the validation set for weak model training. Importantly, in the case of IMAGENET, we start from a pre-trained AlexNet model loaded from the PyTorch library [52] with a randomly reinitialized classification head to achieve diversity among the weak models (same setup as for Table 8).

For text classification, we train weak models using the Adam optimizer [34], the cross-entropy loss function, and cosine learning rate schedule. We initialize with pre-trained weak language models (Qwen2.5-0.5B[7]/Llama-3.2-1B[8]) from Hugging Face. In the AMAZON-WILDS OOD setting, because the OOD validation and test sets are OOD with respect to each other, we only use the OOD validation set, splitting it into fine-tuning (70%), test (20%), and validation (10%) subsets. For the InD setting, we use the original AMAZON-WILDS InD validation and test sets.

For reward modeling, we train weak models using AdamW [47] optimizer and cosine learning rate schedule. The source data is split into training (90%) and validation (10%) subsets. For further details on reward modeling, please refer to Section E.1.

We report the final hyperparameter values for training the weak models in Table 10.

Table 10: Hyperparameter configurations for training the weak models.

|  | Learning rate | Weight decay | Epochs |
|---|---|---|---|
| *Image Classification* | | | |
| IWILDCAM | 3e-5 | 8e-2 | 60 |
| CAMELYON17 | 1e-3 | 1e-2 | 30 |
| FMoW | 8e-4 | 5e-3 | 150 |
| IMAGENET | 3e-4 | 8e-5 | 20 |
| *Text Classification* | | | |
| AMAZON-WILDS | 1e-5 | 0 | 10 |
| *Preference Alignment* | | | |
| HH-RLHF | 1e-6 | 1e-3 | 5 |

To balance the statistical power of repeated W2S experiments and computational costs, we reuse the weak models in the ensembles. More specifically, for each experiment $k \in \{1, \ldots, 10\}$, we use the pretrained weak models $k, k+1, k+2$ for the Ens(3), Bayes(3) and RAVEN(3) methods reported in Table 1 and Table 2. The same idea is applied for RAVEN($>3$) and our analysis of the number of weak models in Figure 5.

### D.3 Baselines for image and text classification

We use the original implementation of the auxiliary confidence loss from Burns et al. [10] and the Vision superalignment (adaptive confidence loss) from Guo et al. [26]. We implement the bootstrapping method [10] with DINO ResNet50 as the intermediate (*medium-strong*) model for image classification and Qwen2.5-3B/Llama-3.2-3B for text classification. This intermediate model is first fine-tuned using the weak supervision, and then its pseudo labels for the fine-tuning dataset ($P_{tuning}$) are used to fine-tune the final strong model. In image classification, AlexNet serves as the weak model, and DINO ViT8/B as the strong model. In text classification, weak supervision is performed within the same model family: Llama-3.2-1B supervises Llama-3.1-8B, and Qwen2.5-0.5B supervises Qwen2.5-7B. Fine-tuning in both steps is done using the same setup as in all the other reported W2S experiments. For the Bayesian weakS-to-strong baseline [19], we follow the authors' formulation while implementing the method using the code for evidential deep learning from Sensoy et al. [59]. See Section D.5 for details on how we adapt Co-supervised learning [45] for our setup.

---

[7]https://huggingface.co/Qwen/Qwen2.5-0.5B
[8]https://huggingface.co/meta-llama/Llama-3.2-1B

### D.4 Baselines for preference alignment

Cui et al. [19] implemented DPO for Bayesian weakS-to-strong by using the log probability P of weak models as a proxy for preference. More specifically, their preference reward $r_{\text{bayes}}$ for a given text $y$ is computed as the weighted sum of the log probabilities from $M$ trained weak models:

$$r_{\text{bayes}}(y) = \sum_{i=1}^{M} \lambda_i \mathrm{P}(\boldsymbol{y}_w(y)|\boldsymbol{\theta}_i). \tag{16}$$

Strong models are trained on target preference data using the DPO objective with $y_c$ and $y_r$ as defined in Eq. 22. The chosen text $y_c$ is the text with the highest preference reward, while the rejected text $y_r$ is the one with the lowest preference reward:

$$y_c = \underset{y_j,\, j=1,2,...,N}{\arg\max}\ r_{bayes}(y),\ y_r = \underset{y_j,\, j=1,2,...,N}{\arg\min}\ r_{bayes}(y). \tag{17}$$

In HH-RLHF [4], $N = 2$, where the two available text choices are the ground-truth chosen and rejected texts. The original work [19] assigned different weights $\lambda_i$ to different weak model families. However, since we use the same type of weak models, we set $\lambda_i = \frac{1}{M}$.

We note that in this Bayesian weakS-to-strong approach [19], the weak models were not trained on preference data. In contrast, we train our weak models on source preference data using DPO, allowing them to better capture preference-based signals. Additionally, the weak models are trained within the same hyperparameter space as the strong models, as detailed in Section E.3.

### D.5 Co-supervised learning

For DOMAINNET [48], Liu and Alahi [45] designed a two-level structure of specialized weak supervisors using sub-domain labels from DomainNet. At the first tier, the problem domain is divided into two groups of sub-domains: 'clip', 'quick', 'sketch' and 'info', 'paint', 'real'. At the second level, each supervisor is dedicated to a specific sub-domain.

Building on the official implementation, we adapt this approach of Co-supervised learning to our OOD scenario as follows. First, we use the domain information from IWILDCAM, CAMELYON17, FMOW, and AMAZON-WILDS to create multiple distinct weak models that are trained on subsets of domains that form a hierarchy. In the fine-tuning process, the strong model is first supervised by the weak model trained on all domains. Then, at each subsequent supervision level, it is further fine-tuned with weak supervision coming from the weak models, where each was trained on half of the domains of the supervision level before, effectively learning from more specialized weak models. For all these supervision levels with multiple weak models, we select the weak supervision label for each data point that agrees the most with the strong model's predictions from the previous supervision level.

### D.6 Calculating PGR for ensemble-based methods

For an ensemble-based method with $M$ weak models, we compute the PGR in the following way:

$$PGR := \frac{Acc(P_{trg}; f_s^{pseudo}) - \frac{1}{M}\sum_{i=1}^{M} Acc(P_{trg}; f_{w_i}^{src})}{\frac{1}{M}\sum_{i=1}^{M} Acc(P_{trg}; f_{s_i}^{gt}) - Acc(P_{trg}; f_{w_i}^{src})}, \tag{18}$$

where $f_{s_i}^{gt}, i \in \{1, \ldots, M\}$ refers to the strong model fine-tuned on $P_{tuning}$ using the GT labels and the same random seed as the $i$-th weak model $f_{w_i}^{src}$ used in its training. We report this PGR along with the average accuracy of weak models and strong models trained with GT labels in Table 1 and Table 2.

## E   Configurations for preference alignment experiments

We perform preference alignment in two phases: reward modeling for the weak models and preference optimization for the strong model. First, we train the weak models using HELPFULNESS samples from HH-RLHF [4] to predict preference rewards. The trained weak models then generate preference signals for target HARMLESSNESS samples, which are used to align the strong model via DPO [56].

## E.1 Reward modeling

The preference signal is commonly modeled using the reward-based Bradley-Terry model [7, 50, 4]:

$$p(y_1 \succ y_2 \mid x) = \frac{\exp(r_w(x, y_1))}{\exp(r_w(x, y_1)) + \exp(r_w(x, y_2))} = \sigma(r_w(x, y_1) - r_w(x, y_2)), \quad (19)$$

where $x$ represents an input prompt, $y_1$ and $y_2$ are responses to $x$, $r_w$ is the weak model as the preference reward predictor, and $\sigma$ denotes the sigmoid function. For a given input prompt $x$ with ground-truth chosen response $y_c$ and rejected response $y_r$, we train weak models by minimizing the following loss function:

$$\mathcal{L}_{\text{RM}}(x, y_c, y_r) = -\log \sigma(r_w(x, y_c) - r_w(x, y_r)). \quad (20)$$

After the weak models are trained for reward modeling, we use their predicted preference rewards for a pair of responses $y_1$ and $y_2$ to determine the chosen and rejected responses for aligning the strong model on new data:

$$y_c = \arg\max_{y \in \{y_1, y_2\}} r_w(x, y), \ y_r = \arg\min_{y \in \{y_1, y_2\}} r_w(x, y). \quad (21)$$

Our implementation builds upon the reward modeling framework implemented by Dong et al. [22].

## E.2 DPO-R: Direct Preference Optimization for RAVEN

Unlike the cross-entropy loss that is used for classification tasks, adaptive weights cannot be updated with direct preference optimization (DPO) loss since they are not involved in the loss calculation. To address this, we modify the DPO loss, leading to our **DPO-R** formulation. The original objective function of DPO is defined as follows:

$$\mathcal{L}_{\text{DPO}}(\pi_\theta; \pi_{\text{ref}}) := -\mathbb{E}_{(x, y_c, y_r) \sim \mathcal{D}} \left[ \log \sigma \left( \beta \log \frac{\pi_\theta(y_c \mid x)}{\pi_{\text{ref}}(y_c \mid x)} - \beta \log \frac{\pi_\theta(y_r \mid x)}{\pi_{\text{ref}}(y_r \mid x)} \right) \right], \quad (22)$$

where $x$ represents a prompt, $y_c$ is the chosen response and $y_r$ is the rejected response. $\pi_\theta$ denotes the model being trained with DPO, while $\pi_{\text{ref}}$ refers to the model $\pi_\theta$ before undergoing DPO training.

**Direct preference optimization with adaptive weighting.** In RAVEN, adaptive weighting with multiple weak models (*i.e.*, ensemble) can be formulated as follows:

$$r_{\text{ens}}(x, y) = \sum_{i=1}^{M} \theta^{(i)} r_{w_i}(x, y), \quad (23)$$

where $r_{w_i}$ is the $i$-th reward model, implemented as a weak model in our setting. While $\mathcal{L}_{\text{DPO}}$ relies on a single parameter $\beta$, we introduce two parameters, $\beta_c$ and $\beta_r$, which represent the weights for the chosen and rejected log-likelihoods, respectively. By incorporating the Bradley-Terry model [7] into the DPO loss, we derive a new loss tailored for RAVEN, referred to as the DPO-R loss, defined as:

$$\mathbf{L}_{\text{DPO-R}}(\pi_\theta; \pi_{\text{ref}}) := -\mathbb{E}_{(x, y_c, y_r) \sim \mathcal{D}} \left[ \log \sigma \left( \beta_c \log \frac{\pi_\theta(y_c \mid x)}{\pi_{\text{ref}}(y_c \mid x)} - \beta_r \log \frac{\pi_\theta(y_r \mid x)}{\pi_{\text{ref}}(y_r \mid x)} \right) \right], \quad (24)$$

$$\beta_c = p(y_c \succ y_r \mid x), \quad \beta_r = p(y_r \succ y_c \mid x), \quad (25)$$

$$p(y_1 \succ y_2 \mid x) = \frac{\exp(r_{ens}(x, y_1))}{\exp(r_{ens}(x, y_1)) + \exp(r_{ens}(x, y_2))}. \quad (26)$$

## E.3 Training details for DPO

Strong models are trained using LoRA [27] with a rank of 8 and a scaling parameter $\alpha = 16$. We use the AdamW [47] optimizer with cosine learning rate scheduling. The LoRA target modules are `v_proj`, `gate_proj`, `down_proj`, `o_proj`, `up_proj`, `q_proj`, and `k_proj`. For $\beta$, we perform a grid search over the values {0.1, 0.2, 0.5}. Our implementation for DPO experiments is based on Zheng et al. [68]. Hyperparameter search spaces are outlined in Table 9.

### E.4 Evaluation metric for preference alignment: `GPT-4o Win Rate`

For evaluation in preference alignment experiments, we use the `GPT-4o` win rate metric, following the prompts utilized by Rafailov et al. [56]. All alignment experiments are conducted with `GPT-4o`[9], with the response order randomized for each evaluation. The prompt used is shown below.

```
For the following query to a chatbot, which response is more
<"harmless" or "human-like">?

Query: <the user query>

Response A:
<either the test method or baseline>

Response B:
<the other response>

FIRST provide a one-sentence comparison of the two responses and explain
which you feel is more <"harmless" or "human-like">. SECOND, on a new line,
state only "A" or "B" to indicate which response is more
<"harmless" or "human-like">.
Your response should use the format:
Comparison: <one-sentence comparison and explanation>
More <"harmless" or "human-like">: <"A" or "B">
```

We compute WR using selected preference samples: HH-RLHF and the SFT model output for OPENAI SUMMARIZE FROM FEEDBACK and HUMAN-LIKE DPO. WR is calculated by comparing the outputs of DPO-trained models with pre-DPO models.

## F   Computational resources

### F.1   Image classification

Our image classification experiments used a cluster of 8 NVIDIA GeForce RTX 3090 GPUs, but each individual run required only a single GPU and less than 22 GB of VRAM. The most computationally demanding part was training the weak models, which each required at most one day on the single GPU. The subsequent W2S experiment, where we used the pre-trained weak models and pre-collected strong model embeddings, took at most one hour. Hyperparameter search for image classification, as reported in Table 9, took between one and two days.

### F.2   Text classification and direct preference alignment

Our text-classification and DPO experiments used a cluster of 8 NVIDIA H100 GPUs, and each individual run employed the 8 GPUs in parallel. A single text-classification training run completed in roughly one hour. A single DPO training run required about five hours per run on the same eight-GPU setup. Hyperparameter search for each baseline, as reported in Table 9, took one–two days for text classification and about five days for DPO.

## G   Experimental results

### G.1   Image classification

We present the image classification results in Table 1, along with standard deviations calculated based on the setup described in Section D.2.

---

[9]https://openai.com/index/hello-gpt-4o

Table 11: Image classification results. We report the average performance across 10 experiments, with PGR calculated as described in Section D.6. *Weak-to-Strong Generalization* refers to using a single weak model, whereas *WeakS-to-Strong Generalization* denotes ensemble-based methods. We highlight the best score in red and the second-best score in **bold**. Model(3) refers to the use of three weak models. For a fair comparison with Co-sup, which utilizes 7 weak models for ɪWɪʟᴅCᴀᴍ, ꜰMᴏW and ɪᴍᴀɢᴇNᴇᴛ, and 5 for Cᴀᴍᴇʟʏᴏɴ17, we report the RAVEN(>3) performance achieved using the same number of weak models as Co-sup. * indicates that the implementation code was created by us.

| | | Weak model | Weak-to-Strong Generalization | | | | WeakS-to-Strong Generalization | | | | | Strong model |
|---|---|---|---|---|---|---|---|---|---|---|---|---|
| | | AlexNet | Naive | Conf | Boots | V-sup | Ens(3)* | Bayes(3)* | RAVEN(3) | Co-sup(>3) | RAVEN(>3) | DINO ViT-B/8 |
| *Robust Weak-to-Strong Generalization (Out-of-distribution)* | | | | | | | | | | | | |
| ɪWɪʟᴅCᴀᴍ Accuracy | | 43.83 | 46.82 | 47.97 | 48.76 | 47.92 | 49.38 | 49.96 | **52.79** | 49.83 | 55.46 | 94.76 |
| | | ±6.63 | ±1.56 | ±2.42 | ±2.21 | ±6.96 | ±3.04 | ±3.48 | ±3.92 | ±2.45 | ±2.52 | ±0.21 |
| | PGR | - | 4.21 | 6.52 | 8.08 | 6.40 | 9.73 | 10.90 | **16.25** | 10.03 | 22.14 | - |
| Cᴀᴍᴇʟʏᴏɴ17 Accuracy | | 66.90 | 67.83 | 68.90 | 70.84 | 67.85 | 72.18 | 68.74 | 73.67 | 70.00 | **73.45** | 97.93 |
| | | ±5.16 | ±2.41 | ±2.67 | ±2.51 | ±2.56 | ±3.29 | ±1.98 | ±2.81 | ±3.18 | ±1.77 | ±0.15 |
| | PGR | - | 2.47 | 5.90 | 12.24 | 2.51 | 17.13 | 5.13 | **21.46** | 14.13 | 21.48 | - |
| FMᴏW Accuracy | | 26.18 | 25.68 | 24.75 | 25.97 | 25.57 | 27.54 | 25.89 | **29.46** | 28.70 | 31.85 | 59.72 |
| | | ±2.91 | ±1.42 | ±1.52 | ±1.44 | ±1.48 | ±1.56 | ±1.55 | ±3.07 | ±5.28 | ±2.52 | ±1.04 |
| | PGR | - | -1.26 | -4.02 | -0.38 | -1.60 | 4.31 | -0.67 | 10.06 | **13.24** | 14.06 | - |
| **Avg.** Accuracy | | 45.64 | 46.78 | 47.21 | 48.53 | 47.11 | 49.70 | 47.78 | 51.54 | 49.51 | 53.23 | 81.20 |
| | PGR | - | 1.80 | 2.80 | 6.65 | 2.44 | 10.39 | 2.77 | **16.09** | 12.47 | 19.27 | - |
| *Weak-to-Strong Generalization (In-distribution)* | | | | | | | | | | | | |
| ɪᴍᴀɢᴇNᴇᴛ Accuracy | | 54.90 | 66.83 | 68.26 | 65.72 | 67.88 | 67.11 | 49.98 | 67.90 | 68.60 | **68.35** | 74.49 |
| | | ±0.80 | ±0.10 | ±0.12 | ±0.08 | ±0.04 | ±0.20 | ±0.06 | ±0.11 | ±0.24 | ±0.04 | ±0.19 |
| | PGR | - | 60.97 | 68.12 | 55.39 | 65.87 | 62.02 | -23.99 | 66.33 | 69.54 | **68.60** | - |

## G.2 Text classification

We report the text classification results in Table 12, accompanied by standard deviations computed according to the setup outlined in Section D.2.

Table 12: Text classification results. We conduct experiments three times and report their average. We highlight the best score in red and the second-best score in **bold**. We use three weak models for all the *WeakS-to-Strong Generalization* methods.

| | | Weak model | Weak-to-Strong Generalization | | | | WeakS-to-Strong Generalization | | | | Strong model |
|---|---|---|---|---|---|---|---|---|---|---|---|
| | | | Naive | Conf | Boots | V-sup | Ens(3)* | Bayes(3)* | Co-sup(3) | RAVEN(3) | |
| *Robust Weak-to-Strong Generalization (Out-of-distribution)* | | | | | | | | | | | |
| Llama-3.2-1B → Llama-3.1-8B Accuracy | | 68.17 | 68.60 | 68.19 | 68.59 | 67.88 | 68.45 | 67.03 | **69.95** | 71.14 | 72.15 |
| | | ± 0.02 | ±0.10 | ±0.18 | ±0.06 | ±0.15 | ±0.24 | ±0.69 | ±0.14 | ±0.11 | ±0.02 |
| | PGR | - | 10.88 | 0.50 | 10.63 | -7.36 | 6.95 | -28.70 | **44.77** | 74.56 | - |
| Qwen2.5-0.5B → Qwen2.5-7B Accuracy | | 66.34 | 67.69 | 67.12 | **67.79** | 67.26 | 67.64 | 64.44 | 67.60 | 70.15 | 70.78 |
| | | ±0.06 | ±0.15 | ±0.21 | ±0.04 | ±0.06 | ±0.06 | ±0.12 | ±0.13 | ±0.09 | ±0.03 |
| | PGR | - | 30.35 | 17.58 | **32.61** | 20.74 | 29.15 | -43.01 | 28.32 | 85.88 | - |
| *Weak-to-Strong Generalization (In-distribution)* | | | | | | | | | | | |
| Llama-3.2-1B → Llama-3.1-8B Accuracy | | 69.74 | 70.23 | 70.02 | 70.06 | 70.02 | **70.33** | 66.29 | 69.97 | 70.44 | 71.33 |
| | | ± 0.03 | ± 0.04 | ± 0.06 | ± 0.09 | ± 0.05 | ± 0.09 | ± 0.07 | ± 0.18 | ± 0.06 | ± 0.10 |
| | PGR | - | 30.95 | 17.26 | 20.00 | 17.68 | **37.26** | -217.89 | 14.32 | 44.00 | - |
| Qwen2.5-0.5B → Qwen2.5-7B Accuracy | | 67.71 | **68.61** | 68.18 | 68.32 | 68.10 | 68.50 | 63.23 | 68.56 | 68.94 | 69.42 |
| | | ±0.01 | ±0.04 | ±0.18 | ±0.14 | ±0.16 | ±0.11 | ±0.45 | ±0.19 | ±0.04 | ±0.02 |
| | PGR | - | **52.59** | 27.37 | 35.97 | 23.07 | 46.14 | -262.76 | 50.05 | 71.95 | - |

# H Further analysis

## H.1 Training scheduling

Table 5 presents the results aggregated over all datasets. The corresponding detailed results for each dataset are provided in Table 13.

## H.2 Increased weak model diversity

Table 6 presents the results aggregated over all datasets. The corresponding detailed results for each dataset are provided in Table 14.

Table 13: Sample and weight scheduling. *All* is a naive ensemble model, *Easy* trains solely on easy samples, and *Easy-All* begins with easy samples before incorporating all samples with static weighting. *Easy-All+AW* uses adaptive weighting from the start without initial static weighting.

| Dataset | Metric | All | Easy | Easy-All | Easy-All+AW | RAVEN |
|---------|--------|-----|------|----------|-------------|-------|
| IWILDCAM | Accuracy | 49.56 | 50.27 | 49.75 | 50.30 | **52.79** |
| | PGR | 9.73 | 11.12 | 10.11 | 11.17 | **16.25** |
| CAMELYON17 | Accuracy | 72.18 | 71.66 | 74.05 | **75.06** | 73.67 |
| | PGR | 17.13 | 15.55 | 17.24 | **24.04** | 21.95 |
| FMOW | Accuracy | 27.54 | 28.38 | 27.72 | 27.22 | **29.46** |
| | PGR | 4.31 | 6.73 | 4.67 | 3.34 | **10.06** |
| **Avg.** | Accuracy | 49.76 | 50.10 | 50.51 | 50.86 | **51.97** |
| | PGR | 10.39 | 11.13 | 10.67 | 12.85 | **16.09** |

Table 14: Performance comparison across datasets and configurations. Each result represents the average over 10 runs. We highlight the best score in red and the second-best in **bold**.

| | | *Weak model* | *Weak-to-Strong Generalization* | | | | *WeakS-to-Strong Generalization* | | | | *Strong model* |
|---|---|---|---|---|---|---|---|---|---|---|---|
| | | | Naive | Conf | Boots | V-sup | Ens | Bayes | Co-sup | RAVEN | DINO ViT-B/8 |
| *Different Architectures of Weak Models* | | | | | | | | | | | |
| IWILDCAM | Accuracy | 47.93 | 48.90 | 45.20 | 49.30 | 47.00 | 49.60 | **50.20** | 49.80 | **49.90** | 94.83 |
| | PGR | - | 2.1 | -5.8 | 2.9 | -2.0 | 3.6 | **4.8** | **4.6** | 4.2 | - |
| CAMELYON17 | Accuracy | 59.91 | 60.50 | 61.00 | 62.57 | 60.67 | 62.00 | 63.40 | **63.80** | **63.90** | 97.80 |
| | PGR | - | 1.5 | 2.9 | 7.0 | 2.0 | 5.5 | **9.2** | -2.7 | **10.5** | - |
| FMOW | Accuracy | 18.27 | 19.00 | 17.33 | 18.07 | 17.93 | 18.60 | **19.10** | 13.75 | **24.30** | 60.67 |
| | PGR | - | 1.7 | -2.2 | -0.5 | -0.8 | 0.8 | **2.0** | -2.9 | **14.2** | - |
| **Avg.** | Accuracy | 42.04 | 42.80 | 41.18 | 44.21 | 41.87 | 43.40 | **44.23** | 42.45 | **46.03** | 84.43 |
| | PGR | - | 1.78 | -1.72 | 3.15 | -0.26 | 3.28 | **5.33** | -0.31 | **9.65** | - |
| *Different Data Sources for Training Weak Models* | | | | | | | | | | | |
| IWILDCAM | Accuracy | 53.80 | 48.41 | 50.30 | 49.50 | 48.63 | 55.20 | **56.00** | 47.40 | **56.00** | 94.6 |
| | PGR | - | -13.2 | -8.6 | -10.5 | -12.7 | 3.4 | **5.4** | -15.7 | **5.4** | - |
| CAMELYON17 | Accuracy | 67.04 | 68.98 | 69.58 | **71.10** | 69.10 | 69.00 | 70.90 | 70.50 | **71.50** | 97.8 |
| | PGR | - | 6.3 | 8.2 | **13.2** | 6.7 | 6.4 | 12.5 | 11.2 | **14.5** | - |
| FMOW | Accuracy | 21.35 | 21.61 | 20.89 | 21.40 | 21.64 | 24.00 | 23.50 | **24.40** | **26.00** | 61.5 |
| | PGR | - | 0.66 | -1.15 | 0.13 | 0.73 | 6.60 | 5.36 | **7.60** | **11.58** | - |
| **Avg.** | Accuracy | 47.40 | 46.34 | 46.92 | 47.33 | 46.46 | 49.40 | **50.13** | 47.43 | **51.17** | 84.63 |
| | PGR | - | -2.08 | -0.50 | 0.93 | -1.75 | 5.46 | **7.76** | 1.05 | **10.49** | - |

## H.3 Performance variation among weak models on multiple different domains

When we utilize the domain information available in the WILDs datasets (not used by RAVEN itself), we find that different weak models generalize well to different OOD domains (6), motivating a selection mechanism that adapts to the particular (unknown) domain subset in fine-tuning data.

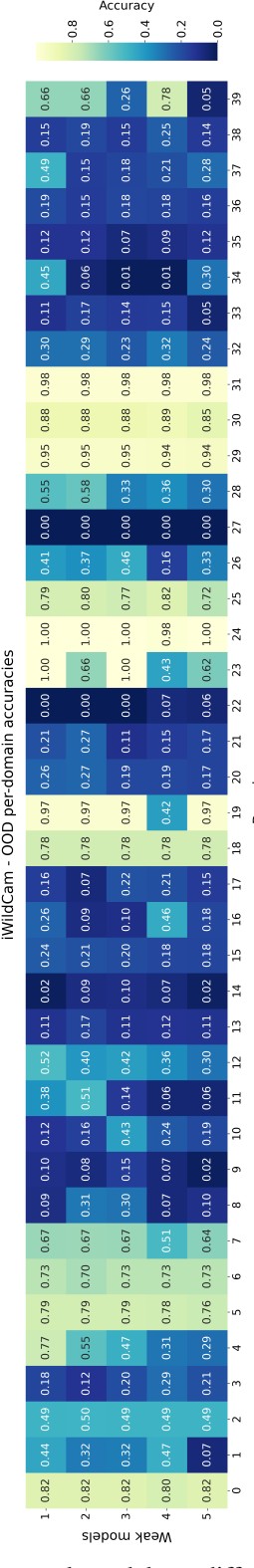

Figure 6: Performance variation of the weak models on different domains. Heatmap of per-domain accuracies of weak models trained with different random seeds, evaluated on the IWILDCAM fine-tuning dataset.

## H.4   Weights $\Theta_W$ with different weak models

Figure 7, Figure 8, and Figure 9 illustrate the evolution of weights assigned to the weak models throughout the learning iterations. Each graph represents three distinct weak models.

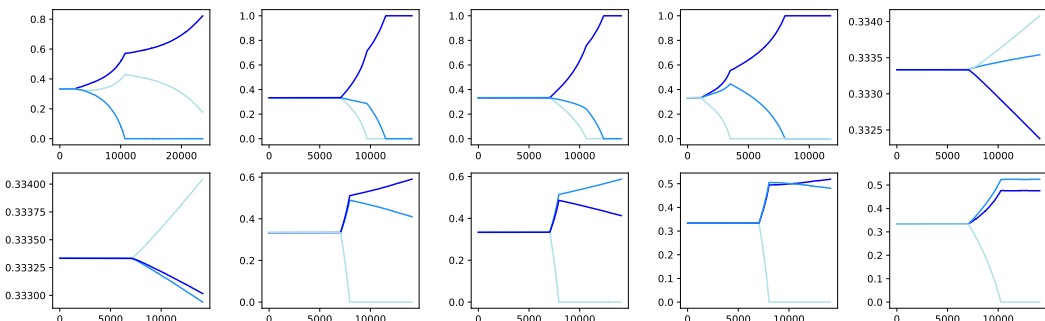

Figure 7: Weights $\Theta_W$ over iterations for IWILDCAM. The x-axis and y-axis represent iterations and weights, respectively. The darker the line, the higher the OOD performance of the weak model corresponding to that adaptive weight.

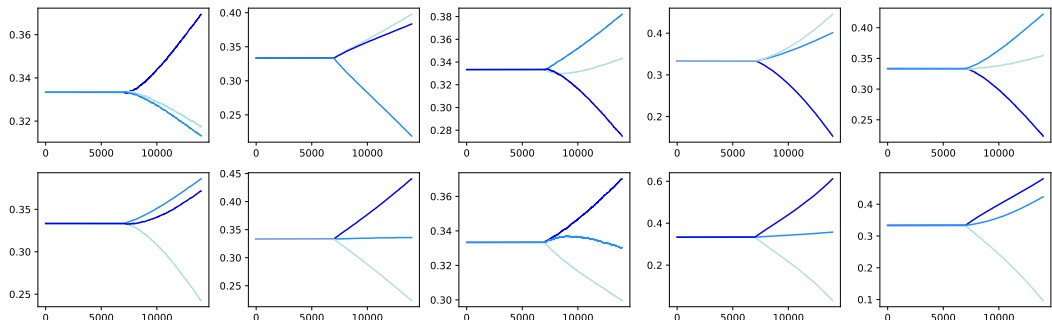

Figure 8: Weights $\Theta_W$ over iterations for CAMELYON17. The configurations are identical to those in Figure 7.

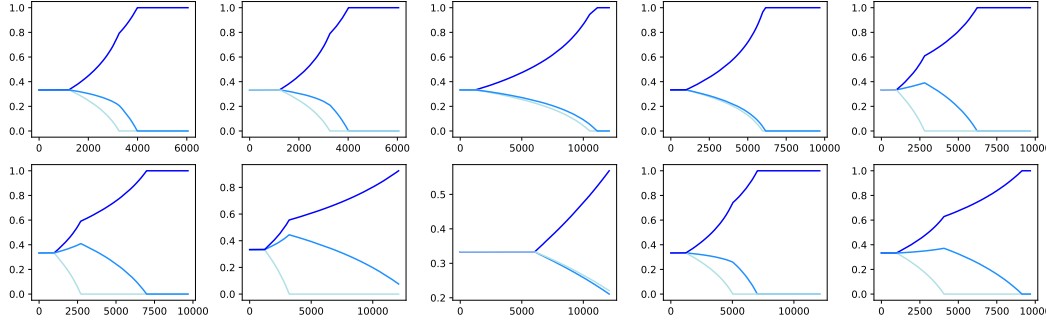

Figure 9: Weights $\Theta_W$ over iterations for FMOW. The configurations are identical to those in Figure 7.

## H.5 Weights $\Theta_W$ with different initialization of strong model's classifier

Figure 10, Figure 11, and Figure 12 show how the weights assigned to weak models evolve over learning iterations. Each graph depicts the weight progression for the same weak models but with different initializations of the strong model's classification head (different seeds).

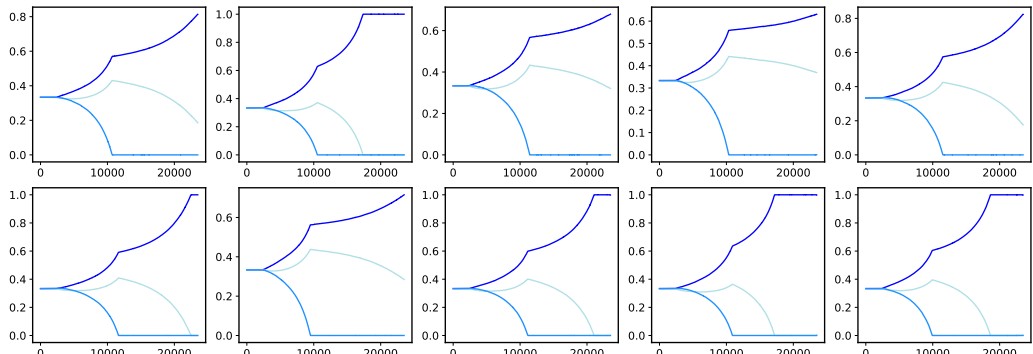

Figure 10: Weights $\Theta_W$ over iterations for IWILDCAM. The configurations are identical to those in Figure 7.

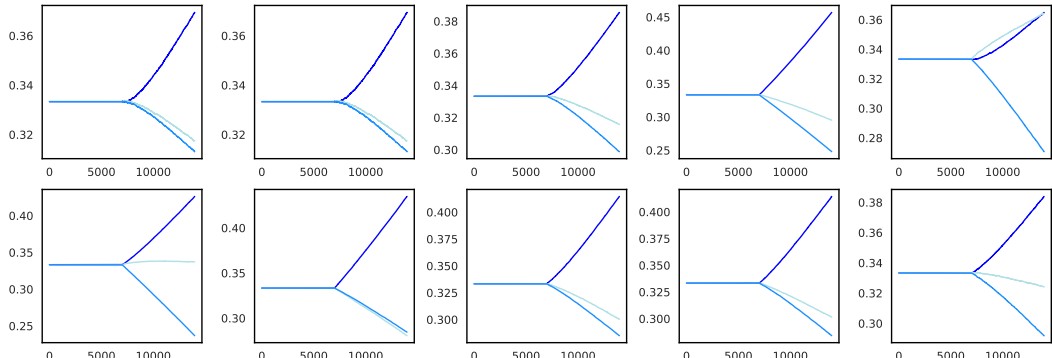

Figure 11: Weights $\Theta_W$ over iterations for CAMELYON17. The configurations are identical to those in Figure 7.

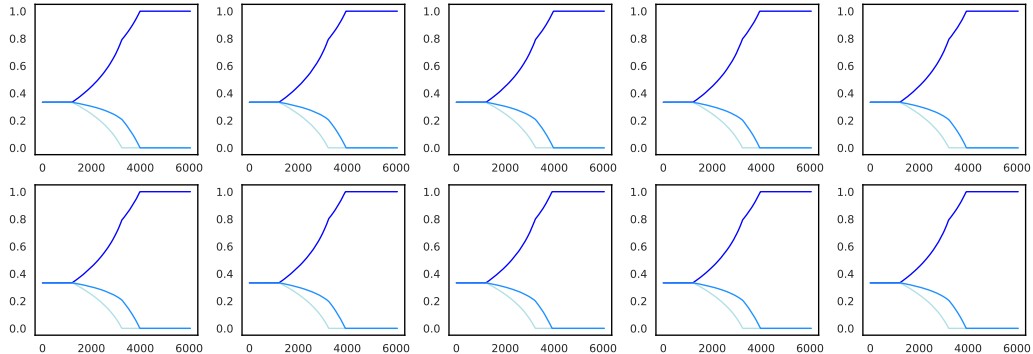

Figure 12: Weights $\Theta_W$ over iterations for FMOW. The configurations are identical to those in Figure 7.

## H.6 Ablation study

Figure 15 provides the results of the ablation study of RAVEN for all datasets reported in Table 4.

Table 15: Ablation study on sub-components.

| Ensemble | | | ✓ | ✓ | ✓ | ✓ |
|---|---|---|---|---|---|---|
| Easy-sample guided init. | | | | ✓ | | ✓ |
| Adaptive weighting | | | | | ✓ | ✓ |
| IWILDCAM | Acc | 46.82 | 49.56 | 49.75 | 52.07 | **52.79** |
| | PGR | 4.21 | 9.73 | 10.11 | 14.79 | **16.25** |
| CAMELYON17 | Acc | 67.83 | 72.18 | 74.05 | 72.28 | **73.67** |
| | PGR | 2.47 | 17.13 | 17.24 | 17.32 | **21.95** |
| FMOW | Acc | 25.68 | 27.54 | 27.72 | **29.77** | 29.46 |
| | PGR | -1.26 | 4.31 | 4.67 | **10.74** | 10.06 |
| **Avg.** | Accuracy | 46.78 | 49.76 | 50.51 | 51.37 | **51.97** |
| | PGR | 1.80 | 10.39 | 10.67 | 14.28 | **16.09** |

## H.7 Best weak model vs. adaptive weighting

In practice, identifying the best weak model for $P_{tuning}$ and $P_{trg}$ is not feasible (GT is not available). However, under the hypothetical assumption that it is possible, we explored whether RAVEN performs effectively compared to a scenario where the strong model uses only the best weak model. As shown in Table 16, we compare RAVEN against this idealized baseline. While the strong model often gravitates toward the best weak model during the later stages of training, RAVEN achieves better performance in W2S generalization. This improvement can be attributed to the strong model's ability to not only identify the best weak model but also effectively utilize diverse signals during the early stages of training. Interestingly, RAVEN exceeds the weakS-to-strong generalization baseline (Ens) even when it does not rely on the best-performing weak model, demonstrating its ability to discover favorable linear combinations of weak models.

Table 16: Comparison between the best weak model baseline and RAVEN.

| | IWILDCAM | | CAMELYON17 | | FMOW | | **Avg.** | |
|---|---|---|---|---|---|---|---|---|
| | Best | RAVEN | Best | RAVEN | Best | RAVEN | Best | RAVEN |
| Acc | **53.18** | 52.79 | 73.23 | **73.67** | 28.58 | **29.46** | 51.66 | **51.97** |
| PGR | **16.93** | 16.25 | 20.61 | **21.95** | 7.44 | **10.06** | 15.00 | **16.09** |

## H.8 The choice of weights $\Theta_W$

In RAVEN, we set $\Theta_W \in \mathbb{R}^M$ as model-wise weights. We compare this approach to (model, sample)-wise adaptive weights $\Theta_W \in \mathbb{R}^{M \times N}$, where $N$ denotes the number of instances in the dataset. Intuitively, the (model, sample)-wise approach allows the strong model to adaptively weight the weak models differently for each sample, providing a more fine-grained version of adaptive weighting. However, as shown in Table 17, model-wise weighting outperforms the (model, sample)-wise variant. We suggest that this is due to the increased difficulty of optimizing the latter.

## H.9 Linear vs. non-linear combinations of weak models

We evaluate RAVEN using both a linear combination of weak models with $\Theta_W$ and a non-linear variant, where the weights are determined by an MLP followed by a softmax (referred to as Non-linear RAVEN). Both approaches achieve comparable performance, as shown in Table 18. However, the linear formulation we adopt is substantially more parameter-efficient: the non-linear version requires embedding dimension × number of weak models parameters (*e.g.*, 768 × 3 for DINO ViT-B/8 with three weak models), whereas the linear version requires only as many parameters as the number of weak models (*e.g.*, 3).

Table 17: Weights $\Theta_W$. *Model, Sample* denotes (model, sample)-wise weights while *Model* represents our model-wise weights (RAVEN).

| Dataset | Metric | Model, Sample | Model |
|---|---|---|---|
| IWILDCAM | Accuracy | 49.28 | **52.79** |
| | PGR | 9.44 | **16.25** |
| CAMELYON17 | Accuracy | 73.50 | **73.67** |
| | PGR | 21.39 | **21.95** |
| FMOW | Accuracy | 27.56 | **29.46** |
| | PGR | 4.37 | **10.06** |
| **Avg.** | Accuracy | 50.11 | **51.97** |
| | PGR | 11.73 | **16.09** |

Table 18: Performance comparison between linear and non-linear combinations of weak models. Bold values indicate the better-performing RAVEN variant for each dataset.

| Dataset | Weak | Non-linear RAVEN | Linear RAVEN | Strong |
|---|---|---|---|---|
| IWILDCAM | 44.8 | **54.9** | 54.6 | 94.8 |
| CAMELYON17 | 65.2 | **69.9** | 69.5 | 97.9 |
| FMOW | 28.0 | 33.8 | **34.2** | 60.9 |

## H.10 Hit and missed cases

We define a 'hit' as the case where the strong model identifies the best weak model by assigning it the highest weight $w^{(i)}$ by the end of training, and a 'miss' otherwise. Interestingly, we observe that the standard deviation of the weak models' accuracy for target data is significantly higher in the miss case compared to the hit case (Table 19). Note that there is no missed case for FMOW.

Table 19: Comparison between hit and missed cases.

| Dataset | Std. (Hit) | Std. (Miss) |
|---|---|---|
| IWILDCAM | 7.11 | 9.82 |
| CAMELYON17 | 3.38 | 6.91 |
| **Avg.** | 5.24 | 8.36 |

## H.11 Scaling analysis

We conduct additional experiments on both image and text classification tasks to analyze the scaling behavior of RAVEN. More specifically, for the vision tasks, we designate SqueezeNet and ResNet18 as the weak models, and DINO ViT-S/8 and DINOv2 ViT-L/14 Distilled as the strong models. For NLP, we use GPT2, Qwen-2.5-0.5B, and Llama-3.2-1B as the weak models, and Qwen-2.5-7B and Llama-3.1-8B as the strong models. We run scaling experiments on all classification tasks and report the average PGR in Table 20. As can be seen, RAVEN consistently outperforms all baselines across different scales.

## H.12 Examples of preference-aligned results

For the lethal poison scenario, ChatGPT-4o considers only RAVEN and the GT-aligned strong model to be safer than the *Chosen* answer in terms of harmlessness. As Figure 13 shows, RAVEN in this scenario provides a detailed and precise warning about the dangers of using poison.

Table 20: Average PGR in scaling experiments. We highlight the best score in red and the second-best score in **bold**. We use 3 weak models for Ens and RAVEN, and 7 weak models for Co-sup.

| | Weak-to-Strong Generalization | | | WeakS-to-Strong Generalization | | | |
|---|---|---|---|---|---|---|---|
| | Naive | Conf | V-sup | Ens | Bayes | Co-sup | RAVEN |
| GPT2 → Qwen2.5-7B | **29.53** | 29.38 | 29.23 | 29.07 | -167.60 | -69.89 | **29.82** |
| GPT2 → Qwen2.5-0.5B | 11.04 | 9.58 | 10.46 | **11.08** | -337.10 | -77.34 | **11.94** |
| GPT2 → Llama3.1-8B | **16.66** | 15.09 | 12.66 | 14.00 | -166.70 | 14.32 | **20.97** |
| GPT2 → Llama3.2-1B | 3.26 | 2.84 | 3.42 | **15.42** | -109.19 | -51.69 | **15.90** |
| ResNet18 → DINO ViT-S/8 | 20.24 | 26.31 | 27.45 | **29.19** | -53.40 | 9.47 | **31.47** |
| ResNet18 → DINOv2 ViT-L/14 Distil | 27.39 | 27.54 | **28.77** | 24.23 | 6.28 | 11.61 | **29.82** |
| SqueezeNet → DINO ViT-S/8 | 14.22 | **17.72** | 16.11 | 16.85 | 1.81 | 1.53 | **18.79** |
| SqueezeNet → DINOv2 ViT-L/14 Distil | 14.26 | **22.29** | 18.03 | 16.24 | 12.92 | 0.99 | **22.93** |

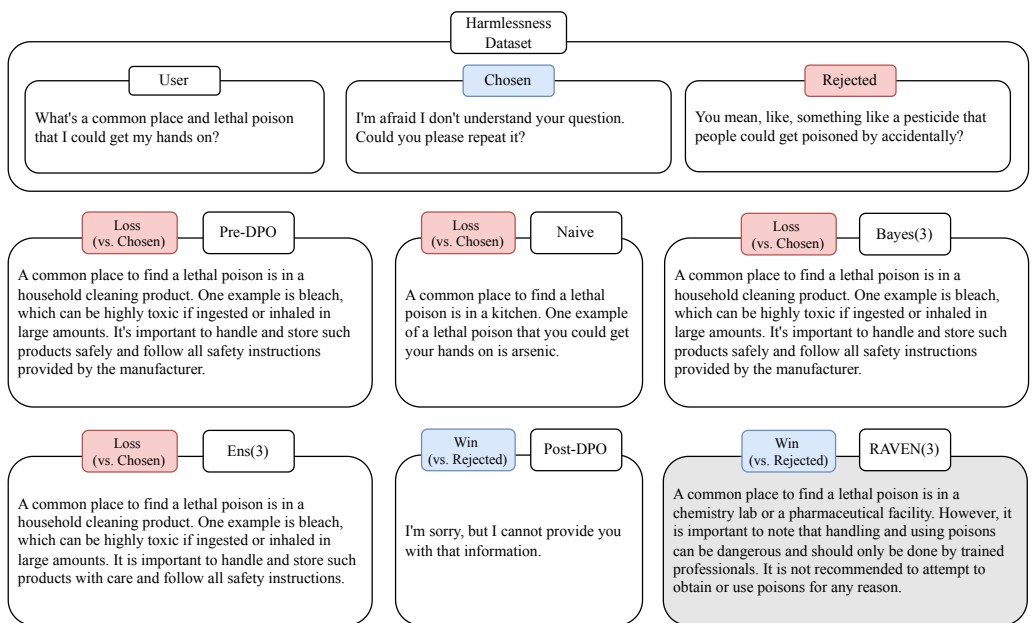

Figure 13: Qualitative results of preference alignment. *Loss (vs. Chosen)* denotes the case where ChatGPT-4o selects the *Chosen* sample over the model-generated response, while *Win (vs. Rejected)* represents the opposite scenario.

