# OpenReview forum: "Weak-to-Strong Generalization under Distribution Shifts"
_NeurIPS.cc/2025/Conference — NeurIPS 2025 poster_

### Official Review · Reviewer_6DE5 · 2025-06-14

**Clarity:** 2
**Significance:** 3
**Originality:** 3
**Rating:** 4
**Confidence:** 3

**Summary:**

The paper introduces RAVEN, a weak-to-strong generalization framework that adaptively weights weak models to reliably supervise a strong model under distribution shifts, achieving significant performance gains across multiple tasks.

**Questions:**

see above

**Ethical Concerns:**

["NO or VERY MINOR ethics concerns only"]

**Final Justification:**

The rebuttal solve my concerns. I will raise my scores.

**Limitations:**

Yes

**Quality:**

3

**Strengths And Weaknesses:**

Pros:

1. The paper proposes RAVEN, a framework that dynamically learns to combine weak models for supervising a stronger model.

2. RAVEN introduces adaptive weighting and easy-sample guided initialization.

3. Extensive experiments on image classification, text classification, and preference alignment show that RAVEN outperforms baselines.

Cons:

1. While the core ideas are sound, the adaptation mechanism (especially how adaptive weights are computed and updated) is difficult to follow. I suggest adding a clear flowchart or algorithmic table that outlines the entire pipeline step-by-step. This would make the framework more accessible to readers unfamiliar with meta-learning or ensemble supervision.

2. The ablation study section could be further strengthened. One useful baseline would be a variant where the weak teachers are used for static weight computation, without adaptive updates. This could isolate the contribution of adaptive weighting more effectively.

3. Table 5 is mentioned in the results but lacks table 5 contents in the paper. This disrupts the flow of the paper and leaves the reader uncertain about the insights it is intended to convey.

---

> ### Author Rebuttal · Authors · 2025-07-31
>
> We thank the reviewer for the comments. We appreciate the reviewer’s recognition that RAVEN’s adaptive weighting and easy-sample guided initialization are well-founded and the core ideas are sound, the extensiveness of our experiments and significant performance gains achieved by RAVEN across multiple tasks.
>
> ---
> **Cons 1.** *While the core ideas are sound, the adaptation mechanism (especially how adaptive weights are computed and updated) is difficult to follow. I suggest adding a clear flowchart or algorithmic table that outlines the entire pipeline step-by-step. This would make the framework more accessible to readers unfamiliar with meta-learning or ensemble supervision.*
>
> **Response.** We thank the reviewer for the suggestion. Since we can not add a figure in the response, we added an algorithmic table below and we will include the figure in the final version of our manuscript.
>
> ---
> ## Algorithm: Robust Adaptive Weighting (RAVEN)
>
> **Require**: Fine-tuning dataset $D$, weak models $\\{ f_{w_i}^{\text{src}} \\}_{i=1}^M$, pretrained strong backbone
> **Ensure**: Trained linear classifier $\Theta_S$
>
> ---
>
> **// Easy-sample guided initialization**
>
> 1. Identify easy samples:
>    $D_{\text{easy}} = \\{ x \in D \mid \arg\max_k f_{w_i}^{\text{src}}(x)[k] = \arg\max_k f_{w_j}^{\text{src}}(x)[k], \text{ for all } i, j \\}$
>
> 2. Initialize weights uniformly:
>    $\theta^{(i)} = \frac{1}{M}$
>
> 3. Train $\Theta_S$ on $D_{\text{easy}}$ by minimizing:
>    $L_{\text{CE}}(f_S(x), f_{\Theta_W}(x))$  where $f_{\Theta_W}(x) = \sum_{i=1}^M \theta^{(i)} f_{w_i}^{\text{src}}(x) \quad$   (Eq. 3)
>
> ---
>
> **// Alternating optimization on the full dataset**
>
> **while not converged do**
>
> 4. Update weights:
>    $\Theta_W \leftarrow \arg\min_{\Theta_W} L_{\text{CE}}(f_S(x), f_{\Theta_W}(x)), x \in D \quad$  (Eq. 4)
>
> 5. Update strong model:
>    $\Theta_S \leftarrow \arg\min_{\Theta_S} L_{\text{CE}}(f_S(x), f_{\Theta_W}(x)), x \in D \quad$  (Eq. 4)
>
> **end while**
>
> **return** $\Theta_S$
>
> ---
>
>
>
> **Cons 2.** *The ablation study section could be further strengthened. One useful baseline would be a variant where the weak teachers are used for static weight computation, without adaptive updates. This could isolate the contribution of adaptive weighting more effectively.*
>
> **Response.** We thank the Reviewer for the suggestion, however, we already incorporated a variant where the weak teachers are used for static weight computation as a baseline that we refer to as Easy-All. The results of comparing static and adaptive weighting are presented in Table 17 in the Appendix G.5. Easy-All and RAVEN share the exact same setup, differing only in the use of static versus adaptive weighting. RAVEN consistently outperforms Easy-All across all datasets.
>
> ---
> **Cons 3.** *Table 5 is mentioned in the results but lacks table 5 contents in the paper. This disrupts the flow of the paper and leaves the reader uncertain about the insights it is intended to convey.*
>
> **Response.** Thank you for pointing out the missing table. We include the missing table below and we will include it in the final version of the manuscript.
>
> Table 5: Sample and weight scheduling. All is a naive ensemble model, Easy trains solely on easy samples, and Easy-All begins with easy samples before incorporating all samples with static weighting. Easy-All+AW adds adaptive weighting from the start without static weighting.
> ||All|Easy|Easy-All|Easy-All+AW|RAVEN|
> |-|-|-|-|-|-|
> |**Acc**|49.76|50.10|50.51|50.86|**51.97**|
> |**PGR**|10.39|11.13|10.67|12.85|**16.09**|

---

> > ### Comment · Reviewer_6DE5 · 2025-08-05
> >
> > Thank you for detailed rebuttal. My concerns have been solved.

---

### Official Review · Reviewer_gvMx · 2025-06-17

**Clarity:** 3
**Significance:** 2
**Originality:** 3
**Rating:** 4
**Confidence:** 3

**Summary:**

This paper introduces RAVEN, a robust weak-to-strong (W2S) generalization framework, to address the performance degradation that occurs when a distribution shift exists between the training data of weak models and the fine-tuning data of a strong model. RAVEN employs an ensemble of weak models and dynamically learns an adaptive weighting scheme to combine their outputs for supervision. The effectiveness of RAVEN is demonstrated through extensive experiments across image classification, text classification, and preference alignment tasks.

**Questions:**

Please refer to the weaknesses.

**Ethical Concerns:**

["NO or VERY MINOR ethics concerns only"]

**Final Justification:**

After checking authors’ responses and other reviewers’ comments, I appreciate that the authors addressed to my questions. Thus, I have decide to raise my rating.

**Limitations:**

yes

**Quality:**

3

**Strengths And Weaknesses:**

Strengths:

    1. This work introduces and formalizes a critical problem in W2S generalization.
    2. RAVEN’s proposed adaptive weighting ensemble is an intuitive and effective solution.
    3. The experiments and analysis are also comprehensive.


Weaknesses:

    1. This paper introduces a novel scenario, named weak-to-strong generalization under distribution shifts. However,  the choice of weak and the strong models is not very representative. For instance, the paper utilizes AlexNet and DINO for image classification, why not evaluate more advanced models like vision-language models?

    2.  The proposed Robust AdaptiVe wEightiNg (RAVEN) approach incorporates adaptive weighting and easy-sample guided initialization. However, both of these techniques are inspired by prior works, so the methodological contribution cannot be considered highly novel. Moreover, the paper does not further explore some aspects, such as whether linear or nonlinear combinations would be more effective in adaptive weighting.

    3. The experimental designs for text classification and preference alignment are not fully convincing, as the performance gap between the weak and strong models is marginal. In the text classification experiments, the weak model underperforms the strong model by only about 4% in accuracy. While in preference alignment, the difference shrinks to less than 2%. A more rigorous experimental setup should employ model pairs with clearly distinguishable capabilities to properly validate the proposed weak-to-strong generalization framework.

---

> ### Author Rebuttal · Authors · 2025-07-31
>
> We thank the reviewer for the suggestions. We are grateful for the reviewer’s recognition that weak-to-strong generalization under distribution shifts, our proposed problem setting, addresses a critical challenge. It is encouraging to know that the reviewer found RAVEN to be an intuitive and effective approach. We also value the reviewer’s positive assessment of the comprehensiveness of our experiments and analysis.
>
> ---
> **Weakness 1.** *This paper introduces a novel scenario, named weak-to-strong generalization under distribution shifts. However,  the choice of weak and the strong models is not very representative. For instance, the paper utilizes AlexNet and DINO for image classification, why not evaluate more advanced models like vision-language models?*
>
> **Response.** We thank the reviewer for the comment. We would first like to emphasize that in addition to AlexNet as a weak model and DINO ViT-B/8 as a strong model, we conducted comprehensive analysis with different (weak, strong) model pairs in the Appendix. In particular, the average PGR across all datasets—including iWildCam, Camelyon17, fMoW, and ImageNet for image classification, and Amazon for text classification—is reported in Table 20 in Appendix G.8. In particular, for image classification we evaluated RAVEN using ResNet18 and SqueezeNet as weak models and DINOv2 ViT-L/14 Distilled and DINO ViT-S/8 as strong models by evaluating all possible combinations. For text classification, we used GPT2, Qwen-2.5-0.5B, and Llama-3.2-1B as the weak models, and Qwen-2.5-7B and Llama-3.1-8B as the strong models
>
> In addition, following the suggestion to include a vision-language model for image classification task, we below show results using a recent vision-language mode (VLM), Qwen2.5-VL [1]. Specifically, we set AlexNet as the weak model and Qwen2.5-VL-3B-Instruct as the strong model. We focused on the iWildCam and Camelyon17 datasets, as Qwen2.5-VL-3B-Instruct performs poorly on the satellite image domain (fMoW), in some cases underperforming even AlexNet (A similar issue was observed with Qwen2.5-VL-7B-Instruct). The results below show that RAVEN remains effective even when supervising a strong VLM. Notably, RAVEN outperforms Co-sup while using only three weak models, whereas Co-sup relies on seven. In response to the Reviewer’s feedback, we will include these results in the final version of the manuscript.
>
> |Dataset|Metric|Weak|Naive|Conf|Boots|V-sup|Bayes(3)|Ens(3)|Co-sup(7)|RAVEN(3)|Strong|
> |-|-|-|-|-|-|-|-|-|-|-|-|
> |iWildCam|Acc|44.7|46.2|47.4|46.1|47.7|47.1|47.6|47.6|**48.7**|86.7|
> |iWildCam|PGR|-|3.5|6.7|3.3|7.2|5.6|6.8|1.2|**9.5**|-|
> |Camelyon17|Acc|65.2|64.5|64.5|64.9|64.5|65.5|64.5|**65.8**|65.7|85.7|
> |Camelyon17|PGR|-|-3.4|-3.7|-1.7|-3.8|1.5|-3.4|-6.7|**2.4**|-|
>
>
> > [1] Bai et al. "Qwen2. 5-vl technical report." arXiv 2025.
>
> ---
> **Weakness 2.** *The proposed Robust AdaptiVe wEightiNg (RAVEN) approach incorporates adaptive weighting and easy-sample guided initialization. However, both of these techniques are inspired by prior works, so the methodological contribution cannot be considered highly novel. Moreover, the paper does not further explore some aspects, such as whether linear or nonlinear combinations would be more effective in adaptive weighting.*
>
> **Response.** Thank you for this comment. We would like to point out that (1) our work is the first to introduce adaptive weighting in weak-to-strong generalization framework where different weights are assigned to weak learners, (2) it is the first work to apply easy-sample guided initialization to weak-to-strong generalization problem, and (3) it is the first work to integrate adaptive weighting with easy-sample guided initialization. Moreover, our work introduces a novel setting weak-to-strong-generlization under distribution shifts motivated by a new finding that weak-to-strong generalization can become infeasible under distribution shifts. Motivated by this finding, our work is the first work to design a robust weak-to-strong pipeline by exploring how strong representations can be leveraged to identify a favorable linear combination of weak models for fine-tuning data annotation. Finally, to extend our framework to the preference alignment setting, we propose **DPO-R**, which addresses the limitation of conventional DPO loss. Specifically, standard DPO cannot support adaptive weighting, as the influence of weak signals is not properly propagated due to its pairwise log-likelihood comparison mechanism (Appendix D.2).
>
> In response to the suggestion regarding nonlinear combinations, we implemented an MLP to determine the weights of the weak models (referred to as Non-linear RAVEN). RAVEN is effective with both a linear combination using $\Theta_W$ and a non-linear MLP followed by softmax and achieves similar performance in both cases. Nevertheless, the linear combination we adopt is more efficient in terms of parameter cost: the non-linear version requires embedding dimension $times$ number of weak models (e.g., 768 × 3 for DINO ViT-B/8 with three weak models), while RAVEN requires only the number of weak models (e.g., 3). We will include these results in the final version of our manuscript.
>
> |Dataset|Weak|Non-linear RAVEN|Linear RAVEN|Strong|
> |-|-|-|-|-|
> |iWildCam|44.8|54.9|54.6|94.8|
> |Camelyon17|65.2|69.9|69.5|97.9|
> |fMoW|28.0|33.8|34.2|60.9|
>
>
> ---
> **Weakness 3.** *The experimental designs for text classification and preference alignment are not fully convincing, as the performance gap between the weak and strong models is marginal. In the text classification experiments, the weak model underperforms the strong model by only about 4% in accuracy. While in preference alignment, the difference shrinks to less than 2%. A more rigorous experimental setup should employ model pairs with clearly distinguishable capabilities to properly validate the proposed weak-to-strong generalization framework.*
>
> **Response.** The difference in performance between weak and strong models is mainly driven by the choice of the models: by adopting stronger models the difference in performance between weak and strong models increases.  To show this, for text classification result, we included experiments with a larger strong model **Qwen-2.5-14B** on the Amazon-Wilds dataset and using Qwen-2.5-0.5B as the weak model (results below). The results show that 1) the gap between a weak and strong model significantly increases, in particular from 4% using Qwen-2.5-7B to **close to 10%** using Qwen-2.5-14B as a strong model, and 2) RAVEN remains the most effective approach in this setting.
>
> Table1: Evaluation from Qwen-2.5-0.5B to Qwen-2.5-14B.
> ||Weak|Naive|Conf|Boots|ACD|Ens|Bayes|CSL|RAVEN|Strong|
> |-|-|-|-|-|-|-|-|-|-|-|
> |Acc|66.34|70.97|70.82|70.00|70.20|70.52|59.07|70.98|**71.34**|75.66|
> |PGR|-|49.7|48.0|39.2|41.3|44.8|-78.0|49.8|**53.6**|-|
>
>
> In addition, we systematically evaluate different combinations of weak and strong model pairs using Qwen-2.5-0.5B and GPT2 as weak models, and Qwen-2.5-7B and Qwen-2.5-14B as strong models. The results consistently show that adopting stronger models leads to larger performance gaps between strong and weak model as expected, and that RAVEN improves performance with the adoption of stronger models.
>
> Table 2: Scaling analysis with Qwen-2.5-0.5B and GPT2 as the weak model. The value shown next to each model indicates its standalone accuracy, while the other values represent RAVEN's weak-to-strong accuracy when transferring from the row model (weak) to the column model (strong).
>
> ||Qwen-2.5-7B **(70.78)**|Qwen-2.5-14B **(75.66)**|
> |-|-|-|
> |**Qwen-0.5B (66.34)**|70.15|71.34|
> |**GPT2 (65.74)**|67.5|70.85|
>
> For preference alignment, we now incorporate the OpenAI Summarize from Feedback Dataset [2] as source data and the Human-Like DPO Dataset [3] as fine-tuning and target data. We use Qwen-2.5-0.5B as the weak model and Qwen-2.5-7B as the strong model. We randomly sample 200 samples to make the target set, and use GPT-4o to compute win rates. The results show that (1) on this dataset, the difference between weak and strong models is **over 25%**, and (2) RAVEN is the best performing model and achieves a 24% PGR improvement over the best alternative baseline Bayes, showing its generalizability to more diverse alignment tasks.
>
> ||Weak|Naive|Ens|Bayes|RAVEN|Strong|
> |-|-|-|-|-|-|-|
> |WR|57.50|67.86|68.09|68.18|**70.37**|83.00|
> |PGR|-|40.6|41.5|41.9|**50.5**|-|
>
> > [2] Stiennon et al. "Learning to summarize with human feedback." NeurIPS 2020.
> > [3] Çalık et al. "Enhancing human-like responses in large language models." arXiv 2025.

---

### Official Review · Reviewer_sL7D · 2025-07-03

**Clarity:** 3
**Significance:** 3
**Originality:** 3
**Rating:** 5
**Confidence:** 3

**Summary:**

This work studies the possibility of training strong models with supervision from weak models where the latter are trained with data from a distribution that is different from the one that the former needs to be trained on. The authors propose to adaptively combine weak models via learnable weights to guide the training of a strong model for this purpose. The experimental results on a number of standard vision and language datasets show strong performance improvements over current SOTA. Experimental results on a number of standard image and text classification, as well as preference alignment tasks show strong performance improvements over current SOTA.

**Questions:**

Please see Weaknesses in the Strengths and Weaknesses section.

**Ethical Concerns:**

["NO or VERY MINOR ethics concerns only"]

**Final Justification:**

**Reponse to initial rebuttal:**
 I thank the authors for the additional experiments and theoretical analyses towards addressing my initial concerns. Below, I provide my response.

While the empirical results are useful in illustrating the applicability of RAVEN under different kinds of distribution shifts, it does not provide any insights into how the nature of the distribution shift should be wrt the weak model, i.e., the issue that a weak model might fail under an arbitrary distribution shift still remains. For this reason, it is important to characterize the family / nature of distribution shifts that a weak model might be able to handle, and those that it cannot. Failure modes are an undeniable component of any system, but it is important to characterize them precisely, without which, reliable applicability is hindered.

My concern is not regarding the scenario when there is sufficient variability among the weak learners - I am convinced that the complementary knowledge across the weak learners would be nothing but beneficial. My concern is regarding the scenario when weak learners do not exhibit sufficient variability. Again, in this case, it is important to understand which types of distribution shifts would lead to sufficient variability among the weak learners, and which ones are more likely to induce a common failure mode for all weak learners.

I thank the authors for providing the theoretical analysis, which addresses my concern about the strong model converging towards the weak model providing the best generalization.

I would therefore continue with my current rating.

**Update after follow-up response from authors:**
I thank the authors for further following up on my feedback about formalizing the nature of distribution shifts that the ensemble of weak models can handle and the characterization of the consequent failure modes. This addresses my concern and I would thus increase my score.

**Limitations:**

Please see Weaknesses in the Strengths and Weaknesses section.

**Paper Formatting Concerns:**

N.A.

**Quality:**

3

**Strengths And Weaknesses:**

## Strengths:

1. The idea of adaptively combining weak models to guide the training of a strong model for weak-to-strong generalization under distribution shifts is a simple and intuitive one.

2. The authors take care to handle practical challenges such as collapse / shortcut learning prevalent in optimization under such scenarios by warmup with easy samples, the predictions on which all weak models agree on, a technique known to be effective in the literature on learning with noisy labels. This drawing of connection between the task of weak-to-strong generalization and learning with noisy labels is an interesting one, and can help further cross-over of ideas between the two areas.

3. The experimental results on a number of standard image and text classification, as well as preference alignment tasks show strong performance improvements over current SOTA.

## Weaknesses:

1. The feasibility of faithfully identifying the correct weak model under distribution shift is not clear. Under arbitrary distribution shifts, weak models may have arbitrarily uncertain predictions [1]. Without specific characterizations of the nature of the distribution shift wrt the weak model, general claims about such identifiability of correct weak models under distribution shift are unreliable.

2. Weak models consistently providing the same predictions may not necessarily be a good indicator of reliability, especially under distribution shift. Since all the weak models in the ensemble are trained on the same source distribution, they are more likely to exhibit the same pattern of mistakes when faced with the same of kind of distribution shift.

3. Remark 1 states that with no additional guidance, the strong model converges to putting the highest amount of weight to the best-performing weak model by the end of training. While this is an interesting observation, it is important to reflect further on this. Why does the strong model necessarily need to converge to the best-performing weak model, in the absence of any information about which weak model is actually good? What characteristic does the best-performing weak model have that drives the optimization of the strong model in its direction?

References:

[1] Ovadia et al., "Can You Trust Your Model’s Uncertainty? Evaluating Predictive Uncertainty Under Dataset Shift", NeurIPS 2019.

---

> ### Author Rebuttal · Authors · 2025-07-31
>
> We thank the reviewer for the thoughtful review. We sincerely appreciate the reviewer's recognition that the problem we introduce, weak-to-strong generalization under distribution shifts, addresses a practical challenge. We're also grateful for the reviewer’s acknowledgment that RAVEN is both intuitive and effective in this context. Moreover, we value the reviewer’s insight that our work opens a new direction by connecting weak-to-strong generalization with learning under noisy labels.
>
> ---
> **Weakness 1.** *The feasibility of faithfully identifying the correct weak model under distribution shift is not clear. Under arbitrary distribution shifts, weak models may have arbitrarily uncertain predictions. Without specific characterizations of the nature of the distribution shift wrt the weak model, general claims about such identifiability of correct weak models under distribution shift are unreliable.*
>
> **Response.** We thank the Reviewer for the comment. Although not explicitly mentioned, in our experiments we tested RAVEN across two different types of distribution shifts: domain shift (iWildCam, Camelyon17, and Anthropic HH-RLHF datasets) and combined domain and subpopulation shifts (fMoW and Amazon datasets). We show that in both of these types of distribution shifts, RAVEN remains effective and outperforms alternative methods. Moreover, we further conducted experiments that involve 19 different synthetic domain shifts on the ImageNet dataset (reported in Table 7 of the Appendix) and showed that RAVEN is consistently effective.
>
> In response, we now additionally conducted experiments under a new distribution type, **spurious correlation**, and report the results below. Following the setup in [1], we used a biased dog-vs-cat image classification dataset with a color-based spurious attribute. Specifically, the source data consists exclusively of bright-colored dogs and dark-colored cats, while the fine-tuning and target data follow a uniform color distribution across both classes. We exclude Co-sup from this experiment, as it requires domain-specific labels that are not available in the spurious correlation benchmark. RAVEN achieves an 8% PGR improvement compared to the best alternative baseline (Boots).  This shows that, while the effectiveness of weak supervision may vary depending on the type and severity of the distribution shift, RAVEN can still identify relatively better weak models that can serve as annotators for the fine-tuning data. In response to the Reviewer’s feedback, we will characterize the type of distribution shift for all datasets in our experiments and include new results.
>
> ||Weak|Naive|Conf|Boots|V-sup|Ens|Bayes|RAVEN|Strong|
> |-|-|-|-|-|-|-|-|-|-|
> |Acc|78.7|83.7|84.5|86.7|86.0|85.7|86.7|**87.4**|99.2|
> |PGR||24.7|28.1|39.3|35.9|34.3|39.1|**42.5**||
>
>
> > [1] Kim et al. "Learning not to learn: Training deep neural networks with biased data." CVPR 2019.
>
> ---
> **Weakness 2.** *Weak models consistently providing the same predictions may not necessarily be a good indicator of reliability, especially under distribution shift. Since all the weak models in the ensemble are trained on the same source distribution, they are more likely to exhibit the same pattern of mistakes when faced with the same of kind of distribution shift.*
>
> **Response.** We thank the Reviewer for the comment. Currently, the diversity of weak learners comes from using different random seeds. This choice is motivated by our finding that weak models exhibit significantly higher variance in the OOD setting compared to the InD setting, 5.43% vs. 0.9% in average accuracy variance across iWildCam, Camelyon17, and fMoW, even when trained on the same source distribution (Figure 3 in the main script). This observation aligns with findings reported in [2].
>
> To further strengthen our results and evaluate RAVEN in the scenario where weak models in the ensemble are not trained on the same distribution, we conducted experiments  by training the weak models on **different data sources** (following the setup in [3], which constructs distinct data sources based on sub-domain). Specifically, we trained seven AlexNet models using different “camera location” domain labels for iWildCam, five AlexNet models using different “hospital” domain labels for Camelyon17, and seven AlexNet models using different “time” domain labels for fMoW. Table 1 shows that RAVEN achieves a 24% PGR improvement compared to the best alternative baseline (Bayes [4]).
>
> Table 1: Different data sources
> ||Alex|Naive|Conf|Boots|V-sup|Bayes|Ens|Co-sup|RAVEN|Strong|
> |-|-|-|-|-|-|-|-|-|-|-|
> |iWildCam|47.2|48.4|50.3|49.5|48.6|56.0|55.2|47.4|56.0|94.6|
> |PGR||2.5|6.5|4.8|3.0|18.5|16.8|0.4|18.5||
> |Camelyon17|67.0|69.0|69.6|71.1|69.1|70.9|69.0|70.5|71.5|97.8|
> |PGR||6.3|8.2|13.2|6.7|12.5|6.4|11.2|14.5||
> |fMoW|21.3|21.6|20.9|21.4|21.6|23.5|24.0|24.4|26.0|61.5|
> |PGR||0.7|-1.2|0.1|0.7|5.4|6.6|7.6|11.6||
> |avg.Acc|45.2|46.3|46.9|47.3|46.5|50.1|49.4|47.4|**51.2**|84.6|
> |avg.PGR||3.2|4.5|6.0|3.5|12.1|9.9|6.4|**14.9**||
>
> In addition, we further evaluated RAVEN in scenario where the variability of weak learners comes from using **different architectures**. We set AlexNet, ResNet18, and SqueezeNet, as weak models to supervise the strong model DINO ViT-B/8. Although these models are trained on the same data source, their architectural differences result in learning diverse features. Table 2 shows that RAVEN achieves a 81% PGR improvement compared to the best alternative baseline (Bayes [4]). Results in Table 1 and Table 2 demonstrate that **RAVEN enables effective weak-to-strong generalization both when weak models have relatively low diversity and when they exhibit substantial diversity.** We will include these results in the final version of our manuscript.
>
> Table 2: Different architectures
> ||W-Avg|Naive|Conf|Boots|V-sup|Bayes|Ens|Co-sup|RAVEN|Strong|
> |-|-|-|-|-|-|-|-|-|-|-|
> |iWildCam|47.9|48.9|45.2|49.3|47.0|50.2|49.6|49.8|49.9|94.8|
> |PGR||2.1|-5.8|2.9|-2.0|4.8|3.6|4.6|4.2||
> |Camelyon17|59.9|60.5|61.0|62.6|60.7|63.4|62.0|63.8|63.9|97.8|
> |PGR||2.9|2.9|7.0|2.0|9.2|5.5|-2.7|10.5||
> |fMoW|18.3|19.0|17.3|18.1|17.9|19.1|18.6|13.8|24.3|60.7|
> |PGR||1.5|-2.2|-0.5|-0.8|2.0|0.8|-2.9|14.2||
> |avg.Acc|42.0|42.8|41.2|44.2|41.9|44.6|43.4|42.5|**46.0**|84.4|
> |avg.PGR||1.8|-1.7|3.1|-0.3|5.3|3.3|-0.3|**9.6**||
>
>
> > [2] Lakshminarayanan et al. "Simple and scalable predictive uncertainty estimation using deep ensembles." NeurIPS 2017.
> > [3] Liu et al. "Co-supervised learning: Improving weak-to-strong generalization with hierarchical mixture of experts." arXiv 2024.
> > [4] Cui et al. "Bayesian weaks-to-strong from text classification to generation." ICLR 2025.
>
> ---
> **Weakness 3.** *Remark 1 states that with no additional guidance, the strong model converges to putting the highest amount of weight to the best-performing weak model by the end of training. While this is an interesting observation, it is important to reflect further on this. Why does the strong model necessarily need to converge to the best-performing weak model, in the absence of any information about which weak model is actually good? What characteristic does the best-performing weak model have that drives the optimization of the strong model in its direction?*
>
> **Response.** To understand why RAVEN often identifies the most suitable weak model for the target data distribution, we begin with the following assumption.
>
> *Assumption 1.* The ensemble of weak models tends to make predictions similar to those of the best-performing weak model:
>
> $\arg\max_{k \in \{1, \dots, C\}} f_{\sum}(x)[k] \approx \arg\max_{k} f^*(x)[k],\quad$ (1)
>
> where  $f_{\sum}(x) := \sum_{i=1}^M f_{w_i}^{\text{src}}(x)$,  and $f^* (x)$ denotes the best-performing weak model. While the ensemble prediction is often close to that of the best model, we assume that $ f^* $ is more accurate and confident.
>
> Let $f_s$ denote the strong model, and define the adaptation loss as:
>
> $L_{\text{adaptation}}(\Theta_S, \Theta_W) :=
> L_{\text{CE}} \left(
> f_s(x), \sum_{i=1}^{M} \theta^{(i)} f_{w_i}^{\text{src}}(x)
> \right) \quad \text{subject to } \sum_{i=1}^{M} \theta^{(i)} = 1, \ \theta^{(i)} \in \Theta_W.\quad$ (2)
>
> This loss is optimized by alternating the following two steps:
>
> **Step 1.** *Optimize* $ \Theta_S $ *with* $ \Theta_W $ *fixed.*
>
> When weights are uniform (i.e., $ \theta^{(i)} = \frac{1}{M} $), the strong model learns to mimic the average behavior of weak models:
>
> $f_s(x) \approx \frac{1}{M} \sum_{i=1}^M f_{w_i}^{\text{src}}(x).\quad$ (3)
>
> **Step 2.** *Optimize* $\Theta_W$ *with* $\Theta_S$ *fixed.*
>
> We rewrite the objective as:
>
> $L_{\text{CE}}(\Theta_W)=\sum_{i=1}^{M} \theta^{(i)} C_i $,  where  $C_i := -𝔼_{x \sim D}$ $\left[ \sum_{k=1}^C f_{w_i}^{\text{src}}(x)[k] \log f_s(x)[k] \right].\quad$ (4)
>
> *Assumption 2.* Each weak model is highly confident in predicting a single class:
>
> $f_{w_i}^{\text{src}}(x)=(\epsilon, \dots, 1-\overline{\epsilon},\dots,\epsilon), \quad \epsilon \ll 1,\quad$ (5)
>
> with the largest value $1-\overline{\epsilon}$ at index $ \hat{k}ᵢ:=\arg\max_k f_{w_i}^{\text{src}}(x)[k] $.
>
> Then,
>
> $\sum_{k=1}^C f_{w_i}^{\text{src}}(x)[k] \log f_s(x)[k]\approx\log f_s(x)[\hat{k}_i]\quad$ (6)
>
> and the objective simplifies to:
>
> $C_i \approx - 𝔼_{x \sim D} \left[ \log f_s(x)[\hat{k}_i] \right].\quad$ (7)
>
> By *Ass.* 1 and Eq. (3), we infer that the best weak model $f^*(x)$ most closely matches the $f_s(x)$ predictions and therefore receives the highest weight—consistent with empirical findings. Notably, this conclusion still holds even when only the best weak model satisfies *Ass.* 2, since the best weak model yields the smallest $C_i$. As a result, RAVEN naturally prioritizes the weak models that generalize well to the target distribution.
>
> Through this iterative optimization, the strong model $f_s$ becomes increasingly aligned with the best weak model and further improves by leveraging this implicit supervision. We will include this in the final version of the manuscript.

---

> > ### Comment · Reviewer_sL7D · 2025-08-05
> > **Response to Rebuttal**
> >
> > I thank the authors for the additional experiments and theoretical analyses towards addressing my initial concerns. Below, I provide my response.
> >
> > 1. While the empirical results are useful in illustrating the applicability of RAVEN under different kinds of distribution shifts, it does not provide any insights into how the nature of the distribution shift should be wrt the weak model, i.e., the issue that a weak model might fail under an arbitrary distribution shift still remains. For this reason, it is important to characterize the family / nature of distribution shifts that a weak model might be able to handle, and those that it cannot. Failure modes are an undeniable component of any system, but it is important to characterize them precisely, without which, reliable applicability is hindered.
> >
> > 2. My concern is not regarding the scenario when there is sufficient variability among the weak learners - I am convinced that the complementary knowledge across the weak learners would be nothing but beneficial. My concern is regarding the scenario when weak learners do not exhibit sufficient variability. Again, in this case, it is important to understand which types of distribution shifts would lead to sufficient variability among the weak learners, and which ones are more likely to induce a common failure mode for all weak learners.
> >
> > 3. I thank the authors for providing the theoretical analysis, which addresses my concern about the strong model converging towards the weak model providing the best generalization.
> >
> > I would therefore continue with my current rating.

---

> > > ### Author Response · Authors · 2025-08-06
> > >
> > > We appreciate the reviewer’s thoughtful feedback, and we are pleased that our theoretical analysis helped address the concern regarding the convergence of the strong model toward the weak model with the best generalization performance.
> > >
> > > Our method focuses on the setting of covariate shift where the input distribution $p(x)$ may differ between the source and fine-tuning data, while the conditional label distribution $p(y \mid x)$ remains unchanged. This is a common setting considered in the distribution shift literature [1, 2]. In this context, the effectiveness of RAVEN depends on the extent to which the weak models’ decision-relevant features remain present in the fine-tuning distribution.
> > >
> > > To make this more precise, let $A$ denote an attribute space, and for each instance $x \in X$, let $a(x) \subseteq A$ be the set of attributes associated with $x$. We define $A_{source} = \bigcup_{x \in D_{source}} a(x)$ and $A_{tuning} = \bigcup_{x \in D_{tuning}} a(x)$, where $D_{source}$ and $D_{tuning}$ are the datasets used to train the weak models and fine-tune the strong model, respectively.
> > >
> > > Each weak model $i$ relies on a subset of attributes $A_{used}^{(i)} \subseteq A_{source}$, representing the features it uses for prediction. A failure mode may arise when, for all weak models $i$, the intersection $A_{used}^{(i)} \cap A_{tuning}$ is empty—that is, none of the features they rely on are present in the fine-tuning data due to an extreme distribution shift. In such cases, we acknowledge that the weak models may not provide informative signals, and RAVEN may not function as intended.
> > >
> > > While such situations are theoretically possible, we did not observe them in our empirical evaluations. In naturally collected real-world data, the source domain typically contains at least some examples that support learning the cues needed for predicting the fine-tuning data [3], making it unlikely that all weak models fail to capture them. **Although RAVEN’s effectiveness may diminish when weak models exhibit limited variability, it can still identify the one that is relatively better aligned with the fine-tuning distribution**, thus providing a useful signal that contributes to improved generalization. We thank the reviewer for the question and we will comment on this in the final version of the manuscript.
> > >
> > > > [1] Taori, Rohan, et al. "Measuring robustness to natural distribution shifts in image classification." NeurIPS 2020.
> > > > [2] Wiles, Olivia, et al. "A fine-grained analysis on distribution shift."  ICLR 2022.
> > > > [3] Koh, Pang Wei, et al. "Wilds: A benchmark of in-the-wild distribution shifts." ICML 2021.

---

> > > > ### Comment · Reviewer_sL7D · 2025-08-06
> > > >
> > > > I thank the authors for further following up on my feedback about formalizing the nature of distribution shifts that the ensemble of weak models can handle and the characterization of the consequent failure modes. This addresses my concern and I would thus increase my score.

---

### Official Review · Reviewer_ZFQR · 2025-07-05

**Clarity:** 3
**Significance:** 3
**Originality:** 3
**Rating:** 5
**Confidence:** 4

**Summary:**

This paper introduces RAVEN, a robust framework for Weak-to-Strong (W2S) Generalization under distribution shifts. W2S generalization is the paradigm where a strong model is trained using supervision from weaker models. The authors observe that naive W2S methods fail under distribution shifts, where the weak model’s training and fine-tuning data distributions differ. RAVEN addresses this by dynamically learning optimal combinations of multiple weak models through adaptive weighting and easy-sample guided initialization. The method is evaluated across image classification, text classification, and preference alignment tasks, showing significant improvements over state-of-the-art baselines, even in out-of-distribution (OOD) settings.

**Questions:**

Suggestions:

1. The current setup varies only random seeds, which doesn’t capture real-world annotator diversity. Using weak models with different architectures, data sources, or noise types would better simulate practical supervision scenarios and strengthen the method’s robustness.

2. The paper lacks theoretical analysis explaining RAVEN’s behavior under distribution shifts. Adding convergence guarantees or formal justification for adaptive weighting would make the approach more rigorous and broadly credible.

3. The current tasks focus on standard classification and narrow preference alignment. Evaluating on more complex domains like medical diagnosis or reasoning tasks would better demonstrate RAVEN’s generalizability.

**Ethical Concerns:**

["NO or VERY MINOR ethics concerns only"]

**Final Justification:**

The author addressed the concerns in the review, but this does not change the paper's overall impact. I will maintain the same score.

**Limitations:**

yes

**Paper Formatting Concerns:**

none.

**Quality:**

2

**Strengths And Weaknesses:**

Strengths
First, the paper addresses a pressing challenge in scalable model supervision: how to train models when high-quality supervision is unavailable or misaligned. By explicitly modeling distribution shifts—a common but under-explored failure mode in W2S settings—it moves the field toward more realistic and robust solutions. Second, the proposed RAVEN framework is methodologically sound, incorporating adaptive weighting and a principled training schedule that includes easy-sample guided initialization. These innovations allow the model to automatically prioritize reliable weak supervision, leading to statistically significant improvements across multiple tasks. Extensive experiments and ablations strongly support the claims, showing consistent gains over diverse and strong baselines.

Weaknesses
One limitation is that the diversity among weak models in RAVEN is achieved only via different random seeds. This simplification might not reflect real-world annotator variance, especially in domains like healthcare or law where supervision quality is tied to domain-specific expertise. Moreover, while the adaptive weighting strategy is powerful, the paper acknowledges challenges in optimizing finer-grained model-sample-specific weights, leaving room for improvement in dynamic supervision. Lastly, the preference alignment experiment, though promising, is narrowly scoped and may not generalize to other types of alignment tasks beyond "helpfulness" and "harmlessness" datasets.

---

> ### Author Rebuttal · Authors · 2025-07-31
>
> We thank the reviewer for thoughtful review and valuable feedback. We are grateful to the reviewer for recognizing that our work addresses a pressing challenge, weak-to-strong generalization under distribution shifts, that will help move the field toward more realistic and robust solutions. We also appreciate the reviewer’s acknowledgement that RAVEN is methodologically sound, demonstrates strong performance and that our extensive experiments and ablations strongly support our claims.
>
> ---
> **Weakness 1, Suggestion 1.** *One limitation is that the diversity among weak models in RAVEN is achieved only via different random seeds. Using weak models with different architectures, data sources, or noise types would better simulate practical supervision scenarios and strengthen the method’s robustness.*
>
> **Response.** Thank you for this important suggestion. In response, we conducted experiments using (1) different architectures and (2) different data sources to enhance the diversity of weak models. For **different architectures**, we used AlexNet, ResNet18, and SqueezeNet, as weak models to supervise the strong model DINO ViT-B/8. Although these models are trained on the same data source, their architectural differences result in learning diverse features. For **different data sources**, we followed the setup in [1], which constructs distinct data splits based on sub-domains. Specifically, we trained seven AlexNet models using different “camera location” domain labels for iWildCam, five AlexNet models using different “hospital” domain labels for Camelyon17, and seven AlexNet models using different “time” domain labels for fMoW.
>
> RAVEN achieves an 81% PGR improvement in the different architectures setting and a 24% PGR improvement in the different data sources setting, compared to the best alternative baselines Bayes. These results demonstrate that RAVEN enables effective weak-to-strong generalization both when weak models have relatively low diversity and when they exhibit substantial diversity. We will include these results in the final version of our manuscript.
> - Different architectures
> ||W-Avg|Naive|Conf|Boots|V-sup|Bayes|Ens|Co-sup|RAVEN|Strong|
> |-|-|-|-|-|-|-|-|-|-|-|
> |iWildCam|47.9|48.9|45.2|49.3|47.0|50.2|49.6|49.8|49.9|94.8|
> |PGR||2.1|-5.8|2.9|-2.0|4.8|3.6|4.6|4.2||
> |Camelyon17|59.9|60.5|61.0|62.6|60.7|63.4|62.0|63.8|63.9|97.8|
> |PGR||2.9|2.9|7.0|2.0|9.2|5.5|-2.7|10.5||
> |fMoW|18.3|19.0|17.3|18.1|17.9|19.1|18.6|13.8|24.3|60.7|
> |PGR||1.5|-2.2|-0.5|-0.8|2.0|0.8|-2.9|14.2||
> |avg.Acc|42.0|42.8|41.2|44.2|41.9|44.6|43.4|42.5|**46.0**|84.4|
> |avg.PGR||1.8|-1.7|3.1|-0.3|5.3|3.3|-0.3|**9.6**||
> - Different data sources
> ||Alex|Naive|Conf|Boots|V-sup|Bayes|Ens|Co-sup|RAVEN|Strong|
> |-|-|-|-|-|-|-|-|-|-|-|
> |iWildCam|47.2|48.4|50.3|49.5|48.6|56.0|55.2|47.4|56.0|94.6|
> |PGR||2.5|6.5|4.8|3.0|18.5|16.8|0.4|18.5||
> |Camelyon17|67.0|69.0|69.6|71.1|69.1|70.9|69.0|70.5|71.5|97.8|
> |PGR||6.3|8.2|13.2|6.7|12.5|6.4|11.2|14.5||
> |fMoW|21.3|21.6|20.9|21.4|21.6|23.5|24.0|24.4|26.0|61.5|
> |PGR||0.7|-1.2|0.1|0.7|5.4|6.6|7.6|11.6||
> |avg.Acc|45.2|46.3|46.9|47.3|46.5|50.1|49.4|47.4|**51.2**|84.6|
> |avg.PGR||3.2|4.5|6.0|3.5|12.1|9.9|6.4|**14.9**||
> > [1] Liu et al. "Co-supervised learning: Improving weak-to-strong generalization with hierarchical mixture of experts." arXiv 2024.
>
> ---
> **Weakness 2.** *While the adaptive weighting strategy is powerful, the paper acknowledges challenges in optimizing finer-grained model-sample-specific weights, leaving room for improvement in dynamic supervision.*
>
> **Response.** **Model-wise weighting (RAVEN) is both more effective and computationally efficient than (model, sample)-wise weighting.** As demonstrated in our experiments (Appendix G.6), RAVEN consistently outperforms the (model, sample)-wise approach. The latter also presents practical limitations, as the number of weights grows with the number of samples—posing scalability challenges, particularly when training strong models on large datasets. Nevertheless, as you noted and we acknowledged in the limitations section, improving the optimization of (model, sample)-wise weighting is a promising direction for future work.
>
> ---
> **Suggestion 2.** *The paper lacks theoretical analysis explaining RAVEN’s behavior under distribution shifts. Adding convergence guarantees or formal justification for adaptive weighting would make the approach more rigorous and broadly credible.*
>
> **Response.** To understand why RAVEN often identifies the most suitable weak model for the target data distribution, we begin with the following assumption.
>
> *Assumption 1.* The ensemble of weak models tends to make predictions similar to those of the best-performing weak model:
>
> $ \arg\max_{k \in \{1, \dots, C\}} f_{\sum}(x)[k] \approx \arg\max_{k} f^*(x)[k],\quad$ (1)
>
> where  $ f_{\sum}(x) := \sum_{i=1}^M f_{w_i}^{\text{src}}(x) $,  and $ f^* (x) $ denotes the best-performing weak model. While the ensemble prediction is often close to that of the best model, we assume that $ f^* $ is more accurate and confident.
>
> Let $ f_s $ denote the strong model, and define the adaptation loss as:
>
> $ L_{\text{adaptation}}(\Theta_S, \Theta_W) :=
> L_{\text{CE}} \left(f_s(x), \sum_{i=1}^{M} \theta^{(i)} f_{w_i}^{\text{src}}(x)
> \right) \quad \text{subject to } \sum_{i=1}^{M} \theta^{(i)} = 1, \ \theta^{(i)} \in \Theta_W.\quad$ (2)
>
> This loss is optimized by alternating the following two steps:
>
> **Step 1.** *Optimize* $\Theta_S $ *with* $ \Theta_W$ *fixed.*
>
> When weights are uniform (i.e., $\theta^{(i)} = \frac{1}{M}$), the strong model learns to mimic the average behavior of weak models:
>
> $f_s(x) \approx \frac{1}{M} \sum_{i=1}^M f_{w_i}^{\text{src}}(x).\quad$ (3)
>
> **Step 2.** *Optimize* $\Theta_W$ *with* $\Theta_S$ *fixed.*
>
> We rewrite the objective as:
>
> $L_{\text{CE}}(\Theta_W)=\sum_{i=1}^{M} \theta^{(i)} C_i $,  where  $C_i := -𝔼_{x \sim D}$ $\left[ \sum_{k=1}^C f_{w_i}^{\text{src}}(x)[k] \log f_s(x)[k]\right].\quad$ (4)
>
> *Assumption 2.* Each weak model is highly confident in predicting a single class:
>
> $f_{w_i}^{\text{src}}(x)=(\epsilon,\dots,1-\overline{\epsilon},\dots,\epsilon),\quad \epsilon \ll1,\quad$ (5)
>
> with the largest value $1-\overline{\epsilon}$ at index  $\hat{k}ᵢ:= \arg\max_k f_{w_i}^{\text{src}}(x)[k] $.
>
> Then,
>
> $\sum_{k=1}^C f_{w_i}^{\text{src}}(x)[k]\log f_s(x)[k]\approx\log f_s(x)[\hat{k}_i]\quad$ (6)
>
> and the objective simplifies to:
>
> $C_i \approx - 𝔼_{x \sim D} \left[ \log f_s(x)[\hat{k}_i] \right].\quad$ (7)
>
> By *Ass.* 1 and Eq. (3), we infer that the best weak model $f^*(x)$ most closely matches the $f_s(x)$ predictions and therefore receives the highest weight—consistent with empirical findings. Notably, this conclusion still holds even when only the best weak model satisfies *Ass.* 2, since the best weak model yields the smallest $C_i$. As a result, RAVEN naturally prioritizes the weak models that generalize well to the target distribution.
>
> Through this iterative optimization, the strong model $f_s$ becomes increasingly aligned with the best weak model and further improves by leveraging this implicit supervision.
>
> ---
> **Weakness 3, Suggestion 3.** *The current tasks focus on standard classification and narrow preference alignment. Evaluating on more complex domains like medical diagnosis or reasoning tasks would better demonstrate RAVEN’s generalizability.*
>
> **Response.** Following this suggestion, we show benefits of RAVEN on the (1) medical diagnosis task, and (2) on the additional preference alignment task.
>
> For **medical domain**, we use three Qwen-2.5-0.5B as the weak models and Meditron-7B and Meditron-70B [2], open-source LLMs adapted to a medical domain, as the strong models. We use MedMCQA [3] (sourced from the All **India** Institute of Medical Sciences) as the source data and MedQA [4] (sourced from **the U.S.** National Medical Board Examination) as the fine-tuning and the target data. Since the datasets come from different countries, there is a domain shift between the source and target. RAVEN achieves a 39% PGR improvement in Qwen-2.5-0.5B → Meditron-7B and an 8% PGR improvement in Qwen-2.5-0.5B → Meditron-70B, compared to the best alternative baselines ACD and Naive, for each task. These results show that RAVEN remains effective even in the complex domain of medical diagnosis.
> - Qwen-2.5-0.5B → Meditron-7B
> ||Weak|Naive|Conf|Boot|ACD|Ens|Bayes|CSL|RAVEN|Strong|
> |-|-|-|-|-|-|-|-|-|-|-|
> |Acc|25.7|26.0|26.3|25.8|25.7|25.7|24.7|25.3|**26.5**|27.6|
> |PGR|-|17.9|30.5|5.3|1.6|-2.6|-52.6|-23.2|**42.6**|-|
> - Qwen-2.5-0.5B → Meditron-70B
> ||Weak|Naive|Conf|Boot|ACD|Ens|Bayes|CSL|RAVEN|Strong|
> |-|-|-|-|-|-|-|-|-|-|-|
> |Acc|25.70|27.57|27.54|27.32|27.26|26.08|20.90|25.22|**27.73**|36.37|
> |PGR|–|17.5|17.2|15.2|14.6|3.6|–45.0|–4.5|**19.0**|–|
>
> In addition, to cover a **broader range of alignment tasks**, we incorporate the OpenAI Summarize from Feedback Dataset [5] as source data and the Human-Like DPO Dataset [6] as fine-tuning and target data. We use Qwen-2.5-0.5B as the weak model and Qwen-2.5-7B as the strong model. We randomly sample 200 samples to make the target set, and use GPT-4o to compute win rates. RAVEN achieves a 24% PGR improvement over the best alternative baseline Bayes, showing its generalizability to more diverse alignment tasks.
>
> ||Weak|Naive|Ens|Bayes|RAVEN|Strong|
> |-|-|-|-|-|-|-|
> |WR|57.50|67.86|68.09|68.18|**70.37**|83.00|
> |PGR||40.6|41.5|41.9|**50.5**||
> > [2] Chen et al. "Meditron-70b: Scaling medical pretraining for large language models." arXiv 2023.
> > [3] Pal et al. "Medmcqa: A large-scale multi-subject multi-choice dataset for medical domain question answering." Conference on health, inference, and learning. PMLR, 2022.
> > [4] Jin, Di, et al. "What disease does this patient have? a large-scale open domain question answering dataset from medical exams." Applied Sciences 2021.
> > [5] Stiennon et al. "Learning to summarize with human feedback." NeurIPS 2020.
> > [6] Çalık et al. "Enhancing human-like responses in large language models." arXiv 2025.

---

### Comment · Area_Chair_6zNC · 2025-07-31
**The reviewer-author discussions are now open.**

The reviewer-author discussions are now open. We kindly ask you to read the author responses and provide your follow-up comments.

---

### Note · Authors · 2025-08-12

We thank all reviewers for their thoughtful reviews and valuable feedback that helped to strengthen our work. We are grateful for their recognition that our proposed problem setting, weak-to-strong generalization under distribution shifts, addresses **a pressing and practical challenge**, moving the field toward more realistic and robust solutions (ZFQR, gvMx). We appreciate the reviewers’ acknowledgement that RAVEN is **methodologically sound, intuitive, and effective**, with well-founded core ideas such as adaptive weighting and easy-sample guided initialization (ZFQR, sL7D, gvMx, 6DE5). We also value their recognition of the **strong performance achieved across multiple tasks** and the comprehensiveness of our experiments and ablations (ZFQR, gvMx, 6DE5), as well as the connection our work draws between weak-to-strong generalization and learning under noisy labels (sL7D).

---

During the rebuttal, we further addressed the reviewers’ concerns through additional **empirical evaluations**, showing RAVEN’s effectiveness with (1) more diverse weak models spanning different architectures and data sources, (2) including a dataset from the medical domain, (3) including additional preference alignment dataset, (4) including challenging weak–strong pairs with large capability gaps (e.g., Qwen-2.5-14B), (5) integration with a state-of-the-art vision–language model, (6) including comparisons between linear and non-linear variants, and (7) scenarios involving spurious correlations. We also provided further **theoretical justification** for RAVEN’s ability to identify the most suitable weak model, thereby ensuring robust performance. We sincerely hope that these additional results and clarifications further highlight the contributions of this work.

---

### Decision · Program_Chairs · 2025-09-17

**Decision:**

Accept (poster)

**Comment:**

This paper addresses the important and practical challenge of weak-to-strong generalization under distribution shifts, a scenario where naive supervision from weaker models fails. The authors propose RAVEN, a methodologically sound framework that dynamically learns to combine an ensemble of weak supervisors through adaptive weighting and an easy-sample guided initialization strategy. Reviewers were uniformly positive about the work, highlighting the novelty and significance of the problem setting, the intuitive and effective nature of the proposed solution, and the comprehensive empirical validation demonstrating strong performance gains across image classification, text classification, and preference alignment tasks. While initial reviews raised some limitations regarding the diversity of the weak models, the scope of the experiments, and the need for further theoretical justification, the authors provided a thorough rebuttal with extensive additional experiments and analysis that successfully addressed these concerns, leading multiple reviewers to raise their scores. The paper's strengths in identifying a critical problem and providing a robust, well-supported solution substantially outweigh any minor initial weaknesses, making it a solid contribution to the field that merits acceptance.